# Characterization and Learning of Causal Graphs with Small Conditioning Sets

**Murat Kocaoglu**
School of Electrical and Computer Engineering
Purdue University
mkocaoglu@purdue.edu

## Abstract

Constraint-based causal discovery algorithms learn part of the causal graph structure by systematically testing conditional independences observed in the data. These algorithms, such as the PC algorithm and its variants, rely on graphical characterizations of the so-called equivalence class of causal graphs proposed by Pearl. However, constraint-based causal discovery algorithms struggle when data is limited since conditional independence tests quickly lose their statistical power, especially when the conditioning set is large. To address this, we propose using conditional independence tests where the size of the conditioning set is upper bounded by some integer $k$ for robust causal discovery. The existing graphical characterizations of the equivalence classes of causal graphs are not applicable when we cannot leverage all the conditional independence statements. We first define the notion of $k$-Markov equivalence: Two causal graphs are $k$-Markov equivalent if they entail the same conditional independence constraints where the conditioning set size is upper bounded by $k$. We propose a novel representation that allows us to graphically characterize $k$-Markov equivalence between two causal graphs. We propose a sound constraint-based algorithm called the $k$-PC algorithm for learning this equivalence class. Finally, we conduct synthetic, and semi-synthetic experiments to demonstrate that the $k$-PC algorithm enables more robust causal discovery in the small sample regime compared to the baseline algorithms.

## 1 Introduction

Causal reasoning is a critical tool for machine learning and artificial intelligence research with benefits ranging from domain adaptation to planning, explainability and fairness [27, 12, 26]. Estimating the effect of an action or an intervention from observational data is called causal inference. A very rich literature of causal inference algorithms have been developed to address this task in the literature [11, 21, 17, 1]. The function that is used to write an interventional distribution in terms of the observational distribution is called the estimand. Estimand depends on the causal relations between the system variables, which are represented in the form of a directed acyclic graph (DAG) called the causal graph. Thus, causal graph is required for solving most causal inference problems.

For small and well-studied systems, it might be possible to construct a causal graph using expert knowledge. However, in modern complex systems with changing structure, we need to learn causal graphs from data. This is called causal discovery. In most domains, we have access to plenty of observational data, but no interventional data. An important task then is to understand how much causal knowledge we can extract from observational data about a system.

The classical approach for addressing this problem is to use the conditional independence (CI) relations in the data to narrow down the space of plausible causal graphs [19, 9]. These are called constraint-based methods. Even though the number of causal graphs that entail the same set of

conditional independence relations are typically exponentially many, we can use a graphical characterization of *equivalence* between causal graphs to compactly represent all of them using a single mixed graph called the *essential graph*. This notion of equivalence is called Markov equivalence.

Even though such causal discovery algorithms are consistent, i.e., they output the correct essential graph in the large sample limit, in practice, they struggle due to finite number of samples since not all CI tests can be performed accurately [16]. For constraint-based algorithms, however, it is important for every test to be accurate since previously learned edge orientations are used to orient more edges due to their sequential nature. Thus, they may output very different causal graphs compared to the ground truth with just a few incorrect CI statements. Despite the efforts to stabilize PC output [2, 5], this fundamental issue still causes problems today. Furthermore, the existing Markov equivalence class characterization and causal discovery algorithms that rely on this characterization, such as the PC/IC algorithms [19, 22], require access to every CI relation, which is a significant practical limitation for causal discovery from data.

There are several alternatives to constraint-based causal discovery. For example, score-based approaches, such as GES [3, 4] optimize a regularized score function by greedily searching over the space of DAGs to output a graph within the Markov equivalence class. A line of works, such as NOTEARS [28] converts the graph learning problem to a continuous optimization problem by converting the acyclicity constraint to a continuous constraint via trace formulation. Note that our goal in this paper is not to beat the state-of-the-art causal discovery algorithm, but provide a theoretical basis to characterize what is learnable on a fundamental level by using conditional independence tests with restricted cardinality conditioning sets.

Variations of the causal discovery problem with limited-size conditioning sets have been considered in the literature. The special case of marginal dependence is considered in [13] and [20]. The most related existing work is [23], where the authors aim learning the graph union of all causal graphs that are consistent with a set of conditional independence statements up to a fixed cardinality conditioning set. They propose a concise algorithm that modifies the steps of PC which they show recovers the union of all equivalent graphs. Similarly in the case with latent variables, AnytimeFCI [18] shows that one can stop FCI algorithm after exhausting all conditional independence tests up to a fixed cardinality conditioning set and the output of the algorithm is still sound for learning parts of the partial ancestral graph (PAG). We propose an alternative route: First, we formally define the equivalence class of causal graphs and propose a graphical object to capture this equivalence class called the $k$-closure graphs. We identify a necessary and sufficient graphical condition to test equivalence between $k$-closure graphs. Finally, we develop a learning algorithm that leverages the representative power of partial ancestral graphs (PAGs), which are typically used in the case with latents, to obtain a provably finer characterization of the set of equivalent causal structures than [23].

In this paper, we propose learning causal graphs using CI tests with bounded conditioning set size. This allows us to ignore CI tests that become unreliable with limited data, and avoid some of the mistakes made by constraint-based causal discovery algorithms. We call CI constraints where the conditioning set size is upper bounded by $k$ as the degree-$k$ CI constraints. We call two causal graphs $k$-Markov equivalent if they entail the same degree-$k$ CI constraints. We propose $k$-closure graphs that entail the same degree-$k$ CI relations as the causal graph, and show that we can characterize $k$-Markov equivalence via Markov equivalence between $k$-closure graphs. We then propose a constraint-based learning algorithm for learning this equivalence class from data, represented by the so-called $k$-essential graph. Finally, we demonstrate that our algorithm can help alleviate the finite-sample issues faced by the PC algorithm, especially while orienting arrowhead marks and for correctly learning the adjacencies in the graph. We also compare our algorithm with NOTEARS [28] and LOCI [23].

## 2    Background

In this section, we give some basic graph terminology as well as background on constraint-based causal discovery: $An(x)$ represents the set of ancestors of $x$ in a given graph that will be clear from context. $De(x)$ similarly represents the descendants of $x$. $Ne(x)$ represents the neighbors of $x$, i.e., the nodes that are adjacent to $x$.

**Definition 2.1.** A path in a causal graph is a sequence of nodes where every pair of consecutive nodes are adjacent in the graph and no node appears more than once.

**Definition 2.2** (Skeleton). The skeleton of a causal graph $D$ is the undirected graph obtained by making every adjacent pair connected via an undirected edge.

**Definition 2.3** ((Unshielded) Collider). A path of three nodes $(a, c, b)$ is a collider if the edges adjacent to $c$ along the path are into $c$. A collider is called an unshielded collider if in addition the endpoints of the path $a, b$ are non-adjacent.

**Definition 2.4** (d-connecting path). A path $p$ is called d-connecting given a set $c$ iff every collider along $p$ is an ancestor of some node in $c$, and every non-collider is not in $c$.

**Definition 2.5** (d-separation). Two nodes $a, b$ on a causal graph $D$ are said to be d-separated given a set $c$ of nodes iff there is no d-connecting path between $a, b$ given $c$, shown by $(a \perp\!\!\!\perp b \,|\, c)_D$, or $a \perp\!\!\!\perp b \,|\, c$ if clear from context.

**Definition 2.6** (Causal Markov assumption). A distribution $p$ is called Markov relative to a graph $D = (V, E)$, if each variable is independent from its non-descendants conditioned on its parents in $D$.

There are local, global and pairwise Markov conditions that can all be shown as a consequence of the Causal Markov condition above [10]. This gives us the following:

**Proposition 2.7** ([10]). *Let $p$ be any joint distribution between varibles on a causal model with the graph $D$. If $(a \perp\!\!\!\perp b \,|\, c)_D$ then $(a \perp\!\!\!\perp b \,|\, c)_p$, i.e., d-separation implies conditional independence under the Causal Markov condition.*

**Definition 2.8** (Markov equivalence). Two causal graphs $D_1, D_2$ are called Markov equivalent, shown by $D_1 \sim D_2$, if they entail the same d-separation constraints.

**Definition 2.9** (Causal Faithfulness assumption). A distribution $p$ is called faithful to a causal graph $D = (V, E)$ iff the following holds $\forall a, b \in V, c \subset V : (a \perp\!\!\!\perp b \,|\, c)_p \Rightarrow (a \perp\!\!\!\perp b \,|\, c)_D$.

Constraint-based causal discovery is not possible without some form of faithfulness assumption, since otherwise CI statements do not inform us of the graph structure. We will later see that restricting ourselves to a small set of CI tests allow us to also weaken the faithfulness assumption we need for our causal discovery algorithm.

**Theorem 2.10** ([22]). *Two DAGs are Markov equivalent if and only if they have the same skeleton and the same unshielded colliders.*

**Definition 2.11** (Degree-$k$ d-separations). The collection of d-separation statements entailed by a DAG where the conditioning set has size of at most $k$ is called degree-$k$ d-separation statements.

## 3 Markov Equivalence of Causal Graphs with Small Conditioning Sets

We are interested in characterizing the collection of causal graphs that entail the same d-separation constraints for all conditioning sets of size up to $k$ for some $k \in \mathbb{N}$. We call this $k$-Markov equivalence:

**Definition 3.1.** Two DAGs $D_1, D_2$ are called $k$-Markov equivalent, shown by $D_1 \sim_k D_2$ if they entail the same d-separation constraints $a \perp\!\!\!\perp b \,|\, c$ for all $c \subset V : |c| \leq k$.

The notion of $k$-Markov equivalence partitions the set of causal graphs into equivalence classes:

**Definition 3.2.** $k$-Markov equivalence class of a DAG $D$ is defined as the set of causal graphs that are $k$-Markov equivalent to $D$, shown by $[D]_k$, i.e., $D' \sim_k D, \forall D' \in [D]_k$.

We first discuss how Verma and Pearl's characterization fails when we cannot test all d-separation statements. Note that if two DAGs have the same skeleton and unshielded colliders, then they entail the same d-separation constraints by Theorem 2.10. Therefore, they also entail the same degree-$k$ d-separation relations. However, the other direction is not true: Two graphs with different unshielded colliders or different skeletons may still entail the same degree-$k$ d-separation relations. This is expected since we are checking less constraints, which increases the size of the equivalence class: More and more graphs become indistinguishable as we do not rely on certain d-separation constraints.

For example, consider the causal graphs in Figure 1 (a, b). In $D_1$, even though $a, b$ are non-adjacent, they cannot be made conditionally independent unless both $c$ and $d$ are conditioned on. Similarly in $D_2$, $c, b$ cannot be made independent unless both $a, d$ are conditioned on. In both graphs, $a, d$ are independent, and become dependent conditioned on $c$ or $b$. Therefore, $D_1, D_2$ are $k$-Markov equivalent for $k = 1$: They entail the same conditional independence relations for conditioning sets

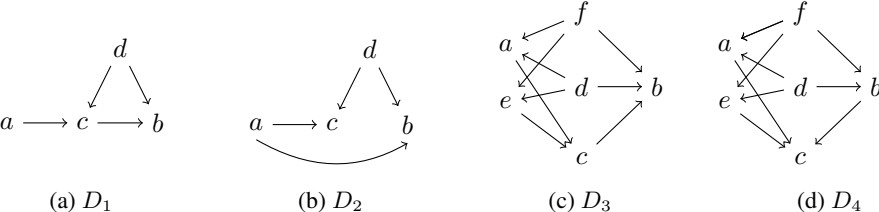

(a) $D_1$        (b) $D_2$        (c) $D_3$        (d) $D_4$

Figure 1: (a), (b): Both $D_1$ and $D_2$ entail the same degree-1 CI relations although they have different skeletons, thus they are in the same $k$-Markov equivalence class. (c), (d): Both $D_3$ and $D_4$ entail the same degree-1 CI relations although they have different unshielded colliders, thus they are in the same $k$-Markov equivalence class.

of size up to 1. Similarly in Figure 1 (c, d), the flipped edge between $c, b$ induces different unshielded colliders in $D_1$ and $D_2$. However, the endpoints of these colliders $(f, c)$, $(d, c)$ and $(e, b)$, $(a, b)$ are all dependent conditioned on subsets of size at most 1. Then it is easy to verify that $D_1, D_2$ entail the same degree-$k$ d-separation relations for $k = 1$ despite having different unshielded colliders.

These examples show that different structures may induce the same degree-$k$ d-separation relations. One might think that, if a collider actually changes the $k$-Markov equivalence class, then perhaps there is hope that a local characterization around all such equivalence class-changing colliders might be possible. However, this is not true. Our example in Section D.1 shows that a local characterization similar to Theorem 2.10 is not possible for $k$-Markov equivalence.

### 3.1 $k$-closure Graphs

Our goal is to come up with a graphical representation of $k$-Markov equivalence class that captures the invariant causal information across equivalent DAGs. This will later be useful for learning identifiable parts of the causal structure from data by CI tests. First, we introduce the notion of a $k$-covered pair.

**Definition 3.3.** Given a DAG $D = (V, E)$ and an integer $k$, a pair of nodes $a, b$ are said to be $k$-covered if $\nexists c \subset V : |c| \leq k$ and $a \perp\!\!\!\perp b \,|\, c$.

Therefore, $k$-covered pairs are pairs of variables that cannot be made independent by conditioning on subsets of size at most $k$. For example, $a, b$ in Figure 1 are $k$-covered for $k = 1$. In any graphical representation of $k$-Markov equivalence class, $k$-covered pairs should be adjacent. This is because it is not always possible to distinguish if they are adjacent or not by degree-$k$ d-separation relations.

We propose $k$-closure graphs as a useful representation to characterize $k$-Markov equivalence class.

**Definition 3.4.** Given a DAG $D = (V, E)$ and an integer $k$, the $k$-closure of $D$ is defined as the graph shown by $\mathcal{C}_k(D)$ that satisfies the following:

1. If: $a, b$ are $k$-covered in $D$
   $i)$ if $a \in An(b)$, then $a \to b$ in $\mathcal{C}_k(D)$,   $ii)$ if $b \in An(a)$, then $a \leftarrow b$ in $\mathcal{C}_k(D)$,
   $iii)$ else $a \leftrightarrow b$ in $\mathcal{C}_k(D)$

2. Else: $a, b$ are non-adjacent in $\mathcal{C}_k(D)$

The definition of $k$-closure trivially implies the following:

**Lemma 3.5.** *Given a DAG $D$ and an integer $k$, the $k$-closure graph $\mathcal{C}_k(D)$ is unique.*

Furthermore, there is a straightforward algorithm one can employ to recover a $k$-closure from the DAG: Make the non-adjacent pairs that cannot be d-separated with conditioning sets of size at most $k$ adjacent according to the definition. Even though this may not be a poly-time operation, it is unavoidable to characterize the $k$-Markov equivalence class.

The following lemma shows that $k$-closure graphs can be used to capture all the conditional independence statements with bounded-size conditioning sets imposed by a DAG.

**Lemma 3.6.** *$k$-closure $\mathcal{C}_k(D)$ of a DAG $D$ entails the same degree-k d-separation statements as the DAG, i.e., $(a \perp\!\!\!\perp b \,|\, c)_D \iff (a \perp\!\!\!\perp b \,|\, c)_{\mathcal{C}_k(D)}, \forall c \subset V : |c| \leq k.$*

*Proof Sketch.* The proof relies on two crucial lemmas. If $a \leftrightarrow b$ in $\mathcal{C}_k(D)$, then in $D$, conditioned on any subset of size at most $k$, there is a d-connecting path with arrowheads on both $a$ and $b$. Similarly, if $a \rightarrow b$ in $\mathcal{C}_k(D)$, then in $D$, conditioned on any subset of size at most $k$ there is a d-connecting path with an arrowhead at $b$. Using these, we can show that any d-connecting path in $\mathcal{C}_k(D)$ implies a d-connecting path in $D$. The other direction is straightforward as the d-connection statements in $D$ hold in $\mathcal{C}_k(D)$ since it is obtained from $D$ by adding edges. Please see Section A.1 for the proof. $\square$

Inspired by ancestral graphs [15], $k$-closure graphs are mixed graphs with directed and bidirected edges, where $k$-covered pairs in the DAG are made adjacent in $\mathcal{C}_k(D)$ based on their ancestrality.

**Definition 3.7.** A mixed graph is a graph that contains directed edges $a \rightarrow b, a \leftarrow b$ or bidirected edges $a \leftrightarrow b$ where every pair of nodes is connected by at most one edge.

Due to bidirected edges, $k$-closure graphs are not DAGs. However, we can show that they are still acyclic. In fact, it is worth noting that $k$-closure graphs are a special class of ancestral graphs [15].

**Lemma 3.8.** *For any DAG $D$, the $k$-closure graph $\mathcal{C}_k(D)$ is a maximal ancestral graph (MAG).*

MAGs have been successfully applied for learning causal graphs with latent variables [15]. Similar to MAGs, $k$-closure graphs are simply graph objects to help us compactly represent the $k$-Markov equivalence class of causal DAGs, rather than expressing the underlying physical system directly. They do, however, represent ancestrality relations between variables by construction.

The relation between $k$-closure graphs and MAGs is not an if and only if relation. In a MAG, one can have a bidirected edge between any pair of nodes as long as it does not create a (almost) directed cycle. This is because they represent latent confounders and one might have latent confounders between any pair of observed nodes. However, bidirected edges in $k$-closure graphs represent $k$-covered pairs and cannot be added arbitrarily. Accordingly, there are MAGs that are not valid $k$-closure graphs. An example is given in Section D.2. We have the following characterization:

**Theorem 3.9.** *A mixed graph $K = (V, E)$ is a $k$-closure graph if and only if it is a maximal ancestral graph and for any bidirected edge $a \leftrightarrow b \in E$ the following is true:*

- $\nexists c \subset V : |c| \leq k, (a \perp\!\!\!\perp b \,|\, c)_{K'}$, *where $K' = (V, E - \{a \leftrightarrow b\})$.*

The Markov equivalence between MAGs has been characterized in the literature. This relies on not only skeletons and unshielded colliders, but also colliders on discriminating paths being identical.

**Definition 3.10.** A path $p = \langle a, z_1, \ldots z_m, u, Y, v \rangle$ is called a discriminating path for $u, Y, v$ if $a, v$ are not adjacent and every vertex $\{z_i\}_i, u$ are colliders on $p$ and parents of $v$.

**Theorem 3.11.** *[[15]] Two MAGs $M_1, M_2$ are Markov equivalent if and only if $i$) They have the same skeleton, $ii$) They have the same unshielded colliders, $iii$) For any node $Y$ for which there is a discriminating path $p$, $Y$ has the same collider status on $p$ in $M_1, M_2$.*

One important observation for our characterization is that Markov equivalence of $k$-closure graphs do not rely on discriminating paths unlike arbitrary ancestral graphs.

**Lemma 3.12.** *Suppose two $k$-closure graphs $K_1, K_2$ have the same skeleton and unshielded colliders. Then along any discriminating path $p$ for a node $Y$, $Y$ has the same collider status in $K_1$ and $K_2$.*

The above lemma shows that discriminating paths, although may exist in $k$-closure graphs, do not alter the equivalence class by themselves. Hence, for the graphical characterization of the equivalence between $k$-closure graphs, we can drop the discriminating path condition.

**Corollary 3.13.** *Two $k$-closure graphs $K_1, K_2$ are Markov equivalent if and only if*

1. *They have the same skeleton and*

2. *They have the same unshielded colliders.*

In the next section, we will prove a $k$-Markov equivalence characterization based on Lemma 3.6, which will later be useful for learning, since we can employ algorithms devised for learning MAGs.

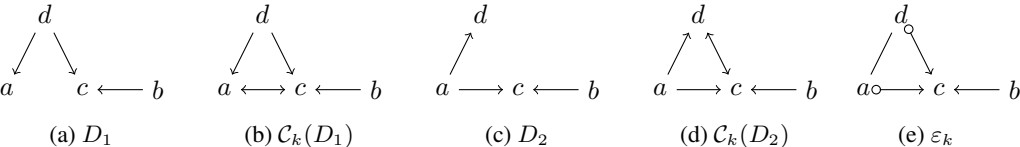

| (a) $D_1$ | (b) $\mathcal{C}_k(D_1)$ | (c) $D_2$ | (d) $\mathcal{C}_k(D_2)$ | (e) $\varepsilon_k$ |

Figure 2: Two $k$-Markov equivalent DAGs for $k = 0$ with the same $k$-essential graph. $D_1 \sim_k D_2$ for $k = 0$. Thus, $\mathcal{C}_k(D_1) \sim \mathcal{C}_k(D_2)$. Thus, they have the same $k$-essential graphs $\varepsilon_k(D_1) = \varepsilon_k(D_2) = \varepsilon_k$, obtained as the edge union of their $k$-closures. Note that there are no Markov equivalent $k$-closures, where $a, d$ are connected with a bidirected edge since removing that edge from the $k$-closure graph would make $a, d$ separable by empty set, which means it would not be a valid $k$-closure graph by Theorem 3.9. Thus, $a, d$ is connected via undirected edge. Similarly, there is no Markov equivalent $k$-closure where $c \leftrightarrow b$ since $c, b$ would not be $k$-covered in any Markov equivalent $k$-closure graph.

## 3.2 $k$-Markov Equivalence

Our main result in this section is the following theorem that characterizes $k$-Markov equivalence:

**Theorem 3.14.** *Two DAGs $D_1, D_2$ are k-Markov equivalent if and only if $\mathcal{C}_k(D_1)$ and $\mathcal{C}_k(D_2)$ are Markov equivalent.*

Thus, $k$-Markov equivalence of two DAGs can be reduced to checking Markov equivalence of their $k$-closures, which can be checked locally using the equivalence condition in Corollary 3.13. Based on this result, it is clear that, just by using conditional independence tests, we can only hope to narrow down our search up to the equivalence class of $k$-closure graphs.

Note that when all the CI tests can be conducted, we can learn the arrowheads and tails that consistently appear in all Markov equivalent DAGs. By operating at the $k$-closure graph-level, we can attain a similar objective and hope to learn the invariant arrowheads and tails.

**Definition 3.15** (Edge union). For our purposes, we define the edge union operation as follows:

$$a \rightarrow b \cup a \leftarrow b = \quad a - b, \quad a \rightarrow b \cup a \leftarrow b \cup a \leftrightarrow b = \quad a \circ\!\!-\!\!\circ b, \quad a \rightarrow b \cup a \leftrightarrow b = \quad a \circ\!\!\rightarrow b$$

**Definition 3.16** ($k$-essential graph). For any DAG $D$, the edge union of all $k$-closure graphs that are Markov equivalent to $\mathcal{C}_k(D)$ is called the $k$-essential graph[1] of $D$, shown by $\varepsilon_k(D)$.

For example, among Markov equivalent $k$-closures, if $a, b$ are adjacent only as $a \rightarrow b$ the $k$-essential graph will have the edge $a \rightarrow b$. Thus, the $k$-essential graph will preserve the invariant arrowhead and tail marks. The difference between $a - b$ and $a \circ\!\!-\!\!\circ b$ is significant from a causal perspective: In the former, two variables cause each other. In the latter, there is some $k$-Markov equivalent DAG where $a, b$ do not cause each other and are simply not separable by conditioning sets of size at most $k$.

For any edge type proposed in the edge union definition, there are relevant instances of $k$-essential graphs. In Figure 2, the edges $a - b$, $a \circ\!\!\rightarrow c$, $c \leftarrow b$ appear in the $k$-essential graph. Similarly, the $k$-essential graph of $D$ in Figure 8 will contain the edge $c \circ\!\!-\!\!\circ d$ since $c \rightarrow d, c \leftarrow d, c \leftrightarrow d$ are all possible edges in different Markov equivalent $k$-closures. Figure 9 in Section D.3 shows an instance where $\leftrightarrow$ appears in the $k$-essential graph. This example also demonstrates that our representation is strictly richer than the graph union that is recovered by LOCI [23], which orients $d \rightarrow c, a \rightarrow c$, and hence cannot distinguish them from the possible edges between $c \leftarrow b$ unlike our representation. Similarly, consider the simple graph $u \rightarrow v$ with $k = 0$. Our representation yields $u - v$ since there is no DAG where $u, v$ are confounded in any way other than the direct edge. In a triangle graph $w \rightarrow u, w \rightarrow v, u \rightarrow v$, our representation recovers $u \circ\!\!-\!\!\circ v$ which shows there are graphs where $u, v$ do not cause each other. LOCI cannot make such distinction as it uses $u - v$ in both graphs.

Therefore, the variant edges that can be arrowheads or tails among different $k$-Markov equivalent graphs are replaced with circles and invariant arrowheads and invariant tails are preserved in the $k$-essential graph. Thus, $k$-essential graph captures the causal information that is preserved across all $k$-Markov equivalent causal graphs. Different from essential graphs where we can test all the

---

[1]The reader will notice $k$-essential graph visually resembles a partial ancestral graph (PAG) [25] more than an essential graph due to the circle marks. Our choice of the name $k$-essential is motivated by the fact that we assume no latent confounders in the system and thus it positions our work better with respect to the related work.

---

**Algorithm 1** $k$-PC Algorithm

---

**Input:** Observational Data, $\mathcal{V}$, $k$, CI Tester $(. \perp\!\!\!\perp . | .)$.

**Step 0:** Initiate a complete graph $K$ between $\mathcal{V}$ with circle edges $o$—$o$.

**Step 1:** Find separating sets $S_{a,b}$ for every pair $a, b \in \mathcal{V}$ by conditioning on subsets of size at most $k$.

**Step 2:** Update $K$ by removing the edges between pairs that are separable.

**Step 3:** Orient unshielded colliders of $K$: For any induced subgraph $a\,o$—$o\,c\,o$—$o\,b$, set $a\,o{\rightarrow} c \leftarrow o\,b$ for any non-adjacent pair $a, b$ where $S_{a,b}$ does not contain $c$.

**Step 4:** $K \leftarrow$ FCI_Orient$(K)$      *# See Algorithm 2 in Section B.*

**Step 5:** For any node $a$ that has no incoming edges (i.e., $a \leftarrow, a \leftrightarrow, a \leftarrow o$) construct the sets $\mathcal{B}, \mathcal{C}$:

$$\mathcal{B} = \{b \in Ne(a) : ao{\rightarrow} b\}, \quad \mathcal{C} = \{c \in Ne(a) : ao\text{—}oc\}$$

and define sets $\mathcal{B}^*$ as the set of nodes that are non-adjacent to any of the nodes in $C$ and $\mathcal{C}^*$ as the set of nodes that are non-adjacent to other nodes in $C$:

$$\mathcal{B}^* = \{b \in \mathcal{B} : b, c \text{ are non-adjacent } \forall c \in C\}, \quad \mathcal{C}^* = \{c' \in \mathcal{C} : c', c \text{ are non-adjacent } \forall c' \neq c, c' \in \mathcal{C}\}$$

- $\mathcal{R}11$: Orient $ao{\rightarrow} b$ as $a \rightarrow b, \forall b \in \mathcal{B}^*$.
- $\mathcal{R}12$: Orient $ao$—$oc$ as $a$—$c, \forall c \in \mathcal{C}^*$.

**Output:** $K$

---

conditional independence relations, existence of an arrow $a \rightarrow b$ in a $k$-essential graph does not mean $a$ causes $b$ in every $k$-Markov equivalent DAG. Rather, it means that in any $k$-Markov equivalent DAG where $a, b$ are adjacent, $a$ causes $b$.

It is worth noting that $k$-essential graphs are in general more informative than partial ancestral graphs (PAGs), which are defined as the edge union of all Markov equivalent MAGs, where the union of $\leftarrow, \rightarrow$ is defined to give $o$—$o$ instead of the undirected edge. Since every Markov equivalent $k$-closure graph is a MAG but not every Markov equivalent MAG is a $k$-closure graph, in general $k$-essential graphs may have invariant arrowheads and tails where PAGs only have circles. We call any graph that contains arrowheads, tails, or circles as edge marks a partial mixed graph (PMG).

**Definition 3.17.** For two partial mixed graphs $A, B$ with the same skeleton, $A$ is said to be a subset of $B$, shown by $A \subseteq B$, iff the following conditions hold for any pair $a, b$

$$1. \ (a\text{—} * b)_A \Leftarrow (a\text{—} * b)_B, \qquad 2. \ (a \leftarrow * b)_A \Leftarrow (a \leftarrow * b)_B. \tag{1}$$

Note that asterisk $*$ stands for a wild-card which can either be an arrowhead, tail, or circle. According to the definition, any circle mark in $A$ is also a circle mark in $B$. We have the following lemma that relates $k$-essential graphs to partial ancestral graphs of $k$-closures.

**Lemma 3.18.** $\varepsilon_k(D) \subseteq PAG(\mathcal{C}_k(D))$.

*Proof.* By Theorem 3.9, every $k$-closure graph is a MAG whereas every MAG is not a $k$-closure graph. Thus the set of Markov equivalent $k$-closure graphs form a subset of the set of Markov equivalent MAGs. Thus, the arrowheads and tails that appear in all Markov equivalent MAGs must also appear in all the Markov equivalent $k$-closure graphs. The result follows from Definition 3.17. $\qquad\square$

In the next section, we propose a sound algorithm for learning $k$-essential graphs from data.

## 4  Learning $k$-essential Graphs

Constraint-based causal discovery algorithms use CI tests to extract all the causal knowledge that can be identified from data. In the previous section, we proposed a compact graphical characterization of what is learnable from such statistical tests. In this section, we propose a constraint-based learning algorithm. Since $k$-closure graphs are a special class of maximal ancestral graphs, we can use FCI algorithm that is devised for learning the invariant arrowheads and tails of a maximal ancestral graph.

FCI algorithm is sound and complete for learning PAGs: It can recover all invariant arrowheads and all invariant tails. One might think that FCI algorithm may also be sound and complete for our task. However, this is not true. Although sound, FCI is not complete for learning $k$-essential graphs.[2]

For soundness, we need the following lemma, which shows that discriminating paths do not carry extra information about the underlying causal structure.

**Lemma 4.1.** *In any $k$-closure graph, if there is a discriminating path $p$ for $\langle u, Y, v \rangle$ and $u \leftrightarrow Y \leftarrow\!\ast v$ is a collider along $p$, then the orientations $u\ast\!\!\rightarrow Y$ and $Y \leftarrow\!\ast v$ can be learned by first finding all unshielded colliders, and then applying the orientation rules $\mathcal{R}1$ and $\mathcal{R}2$ of $FCI$.*

The above lemma shows that the colliders that are part of discriminating paths can be learned by simply orienting the unshielded colliders and then applying the orientation rules of FCI. This is useful since we do not need to search for colliders along discriminating paths during learning.

Our constraint-based causal discovery algorithm, called $k$-PC , is given in Algorithm 1. It uses FCI Orient algorithm in Section B, with two extra rules specific for learning $k$-essential graph.

**Corollary 4.2.** *$k$-PC without Step 5 is sound and complete for learning $PAG(\mathcal{C}_k(D))$ of any DAG $D$.*

*Proof.* By Lemma A.18, any non-adjacent pair are separable by a set of size at most $k$. Thus, a valid separating set for any non-adjacent pair will be found in Step 1, and will be used to learn the skeleton in Step 2, and to orient all unshielded colliders in Step 3. [25] proved arrowhead and tail completeness of FCI with orientation rules $\mathcal{R}1$ to $\mathcal{R}10$. By Lemma 4.1, colliders on discriminating paths will be oriented at the end of Step 4. Thus, $\mathcal{R}4$ that is concerned with discriminating path colliders is not applicable. Similarly, $\mathcal{R}5, \mathcal{R}6, \mathcal{R}7$ are only applicable in graphs with selection bias. Thus, they are not applicable. Since the rules that we omit are never applicable during the execution of the algorithm, Step 4 correctly returns the $PAG(\mathcal{C}_k(D))$ since the algorithm at that point is identical to the FCI algorithm for learning PAGs. $\square$

**Definition 4.3.** A distribution $p$ is said to be $k$-faithful to a causal graph $D = (V, E)$, iff $(a \perp\!\!\!\perp b \,|\, c)_p$ implies $(a \perp\!\!\!\perp b \,|\, c)_D$ for all $c \subset V : |c| \leq k$.

**Theorem 4.4.** *$k$-PC algorithm is sound for learning $k$-essential graph given a conditional independence oracle under the causal Markov and $k$-faithfulness assumptions, i.e., if $k$-PC returns $K$, we have $\varepsilon_k(D) \subseteq K \subseteq PAG(\mathcal{C}_k(D))$*

*Proof Sketch.* From Corollary 4.2, $K$ obtained at the end of Step 4 is $PAG(\mathcal{C}_k(D))$ and by Lemma 3.18 we know $\varepsilon_k(D) \subseteq PAG(\mathcal{C}_k(D))$, i.e., every edge and tail orientation of $K$ is consistent with $\varepsilon_k(D)$. Thus, we only need to show that orientation rules $\mathcal{R}11, \mathcal{R}12$ are sound, i.e., there exists no $k$-closure graph in the Markov equivalence class that is inconsistent with these orientation rules. By Lemma A.2, we know that if $a \leftrightarrow x$ for some $x$, then conditioned on any subset of size $k$, there exists a d-connecting path that starts with arrow at $a$ and $x$. This means that, irrespective of $k$, $a$ must have some incoming edge. The rules use this fact together with the fact that if there were two incoming edges from two non-adjacent neighbors, this would create an unshielded collider, which would change the Markov equivalence class. Please see Section A.8 for the proof. $\square$

For two sample runs of the algorithm, please see Figure 5 and Figure 6 in Section C of Appendix. Note also that our algorithm can be seen as an improved version of AnytimeFCI algorithm [18] for causally sufficient systems, which the author shows can be stopped once every conditioning set of size at most $k$ are tested. While FCI aims at learning arbitrary PAGs partially, $k$-PC learns $k$-closure graphs for causally sufficient systems. This allows extracting more causal information, evidenced by the additional orientation rules employed by $k$-PC . For more details on the differences and an example where AnytimeFCI is less informative than $k$-PC , please see Section D.7.

**Computational Complexity.** Suppose we are given two causal graphs. Our characterization gives an algorithmic way of testing $k$-Markov equivalence: Construct the $k$-closure graphs given the two DAGs and check whether they are Markov equivalent. The step to make two variables adjacent requires one to loop through all conditioning sets of size at most $k$. This would take $\mathcal{O}(n^k)$. This is the main time-consuming step. Afterward, one can test Markov equivalence using the existing

---

[2]It is worth noting that FCI uses undirected edges to represent selection bias. We use undirected edges for a different purpose. Thus, orientation rules of FCI aimed at orienting undirected edges should not be used here.

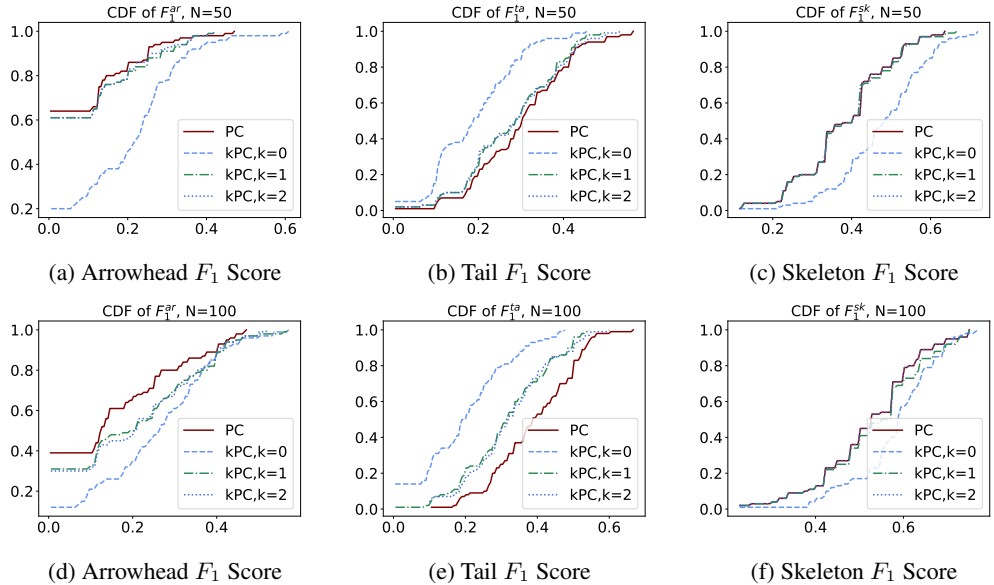

Figure 3: Empirical cumulative distribution function of various $F_1$ scores on 100 random DAGs on 10 nodes. For each DAG, conditional probability tables are independently and uniformly randomly filled from the corresponding probability simplex. Three datasets are sampled per model instance. The lower the curve the better. The maximum number of edges is 15. $N$ is the number of data samples.

approaches. For example, [7] show that this can be done in $\mathcal{O}(ne^2 + n^2 e)$ time, where $e$ is the number of edges. Thus, the overall algorithm will indeed be polynomial-time when $k = \mathcal{O}(1)$.

The complexity of the learning algorithm, $k$-PC , will be similar to Anytime FCI, an early-stopped version of FCI [18]. Although this is more complicated and would depend on other parameters in the graph, such as primitive inducing paths, and thus the number and location of unobserved confounders, we can also roughly bound this by $\mathcal{O}(n^{k+2})$ since for any pair, we will not be searching for separating sets beyond the $\mathcal{O}(n^k)$ subsets of size at most $k$. Further runtime improvements, such as RFCI by [6] might be possible, but this requires a further understanding of the structure of $k$-closure graphs.

## 5 Experiments

### 5.1 Synthetic Data

In this section, we test our algorithm against the stable version of PC algorithm in causal-learn package. We randomly sample DAGs using the pyAgrum package. Variation of this experiment with a different sparsity level can be found in Section E.3 and gives similar results. All variables are binary with conditional probability tables filled randomly uniformly from the corresponding dimensional probability simplex, except the linear SCM experiments for comparing with NOTEARS. We observed similar results with a larger number of states, which are not reported. We report the $F_1$ scores for identifying arrowheads, tails and skeleton correctly. The results are given in Figure 3. We observe that $k$-PC provides significant improvement to the arrowhead $F_1$ score in the small-sample regime at little cost to the tail score. $k$-PC also helps improve skeleton discovery. Similar improvements are seen even for 10 samples whereas the advantage disappears for more than 500 samples (see Section E.3) since PC becomes reliable enough and $k$-PC does not use some of the informative CI tests in that high-sample regime. We report combined metrics such as the sum of all $F_1$ scores in Section E.4.

In Section E.5, we compare with the conservative version of PC [14] and observe similar results. In Section E.2 we generate linear SCMs randomly and compare the performance of our algorithm with NOTEARS in addition to PC. Even though NOTEARS performs slightly better than PC in general, $k$-PC maintains an advantage over both in the small-sample regime. In Section E.1, we compare our algorithm with LOCI [23]. As expected, the two algorithms perform similarly. We observe that our algorithm shows better arrowhead and skeleton $F_1$ score performance. As a side note, we show that

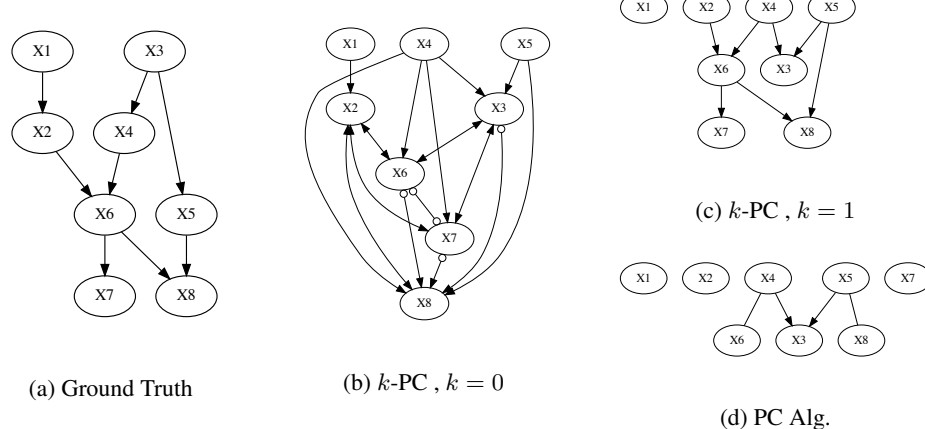

(a) Ground Truth

(b) $k$-PC , $k = 0$

(c) $k$-PC , $k = 1$

(d) PC Alg.

Figure 4: Causal discovery on *Asia* dataset using $500$ random samples. $k$-PC can learn the approximate causal order and some edges accurately, whereas PC outputs a very sparse graph since it conditions on subsets of size 2, which over-sparsifies the graph. For example, $k$-PC with $k = 1$ can recover that $X_2$ is an ancestor of $X_6$, which is an ancestor of $X_7$, whereas PC cannot.

$k$-PC output is at least as informative as LOCI in Section D.6. See Section D for further discussions. The Python code is provided at `https://github.com/CausalML-Lab/kPC`.

## 5.2 Semi-synthetic Data

We test our algorithm on the semi-synthetic *Asia* dataset from bnlearn repository and compare it with PC algorithm. The dataset contains 8 binary variables. We randomly sample 500 datapoints from the observational distribution and run PC as well as $k$-PC for $k = 0$ and $k = 1$. $k$-PC for $k = 0$ correctly gets some of the causal and ancestral relations (Figure 4). PC, on the other hand, recovers a very sparse graph as it utilizes unreliable conditional independence tests with conditioning size of 2. $k$-PC with $k = 1$ recovers a graph in-between the two, still more informative about the structure than PC.

## 6  Discussion

A limitation of the method is that it assumes all independence statements up to degree $k$ can be tested for some $k$. In some cases, the set of available, or the set of reliable CI statements might not have such a structure. Our method is not directly applicable in such scenarios. Another limitation is that we assume that we can run CI tests for the tests that are deemed reliable. This is a non-trivial problem when the data is non IID, such as time-series data. We also make some other usual assumptions that are commonly made in causal discovery, such as acyclicity.

For more remarks, such as a demonstration of incompleteness of $k$-PC , please see Section D.

## 7  Conclusion

We proposed a new notion of equivalence between causal graphs called the $k$-Markov equivalence. This new equivalence allows us to learn parts of the causal structure from data without relying on CI tests with large conditioning sets that tend to be unreliable. We showed that our learning algorithm is sound, and demonstrated that it can help correct some errors induced by other algorithms in practice.

## Acknowledgements

This research has been supported in part by NSF CAREER 2239375. We would like to thank Marcel Wienöbst for his insightful comments and suggestions on an earlier version of the manuscript that led to a more thorough comparison with LOCI [23]. We would like to also thank the anonymous reviewers for their constructive comments and feedback during the review process.

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

# Appendix

## A    Proofs

In this section, we provide proofs for the lemmas and theorems in the paper. We also present FCI orientation rules for completeness, demonstrate why $k$-PC is incomplete, and give two sample runs of the $k$-PC algorithm. Each subsection starts from a new page to clearly separate the proofs of different lemmas/theorems.

### A.1    Proof of Lemma 3.6

We will crucially use the following three lemmas to prove our main results. We say a collider $\langle u, v, w \rangle$ is closed, or blocks the path in context if no node in $De(v)$ is in the conditioning set. Similarly, a path is called closed if it is not d-connecting, and open otherwise.

**Lemma A.1.** *Consider a DAG where $X \notin An(Y)$. Suppose there is a d-connecting path $p$ between $X, Y$ given $T$ that starts with an arrow out of $X$.*

1. *There is at least one collider along $p$.*

2. *Let $K$ be the collider closest to $X$ on $p$. Then conditioning on $T' = T - De(K)$ instead of $T$ does not introduce new d-connecting paths that start with an arrowhead at $X$.*

*Proof.* 1. Any path that starts with $X \to \ldots$ must either be directed, or there must be at least one collider along the path. Since the path is between $X, Y$ and $X \notin An(Y)$, it must be that the path has at least one collider on it.

2. First note that without loss of generality, $p$ has the following form for some integer $m \geq 0$ ($m = 0$ means $X \to K$):
$$X \to U_1 \to U_2 \ldots \to U_m \to K \leftarrow \ldots Y. \tag{2}$$

Suppose for the sake of contradiction that conditioned on $T'$ there is a new d-connecting path $q$ that starts with an arrowhead into $X$. $q$ was clearly closed conditioned on $T$ and become open by us removing nodes from the conditioning set $T$. This can only happen if we removed some node from $T$ that is a non-collider along $q$. Consider the non-collider we removed that was closest to $X$, call this $M$. Thus, we have the path $q$ that looks like this:

$$X \leftarrow W \ldots M \ldots Y, \tag{3}$$

for some $W$, where $M$ is a non-collider on this path and is in $De(K)$.

We observe that the subpath between $M$ and $X$ cannot be directed from $M$ to $X$. Because this would create the following cycle:

$$M \to \ldots \to X \to U_1 \ldots U_m \to K \to \ldots \to M. \tag{4}$$

Thus a closer look at the path $q$ reveals the following structure for some integer $m'$ and node $V$:

$$X \leftarrow W_1 \leftarrow W_2 \ldots \leftarrow W_{m'} \to V \ldots M \ldots Y \tag{5}$$

We will consider the following two cases: The edge adjacent to $M$ along the subpath between $W_{m'}$ and $M$ is a tail or an arrowhead.

**Suppose the edge adjacent to $M$ along the subpath between $W_{m'}$ and $M$ is a tail:** This means there is at least one collider between $W_{m'}$ and $M$. Any such collider must be active since $q$ is active given $T'$. Consider the collider that is closest to $M$. Since it is active, this collider must be an ancestor of $T'$. However, observe that $K$ is an ancestor of this collider which implies that $K$ is an ancestor of $T'$ as well. However, we obtained $T'$ by removing all descendants of $K$ from $T$, which is a contradiction.

**This establishes that the edge adjacent to $M$ along the subpath between $W_{m'}$ and $M$ is an arrowhead.** Thus, this reveals the following structure for $q$:

$$X \leftarrow W_1 \leftarrow W_2 \ldots \leftarrow W_{m'} \to \ldots \to M \ldots Y \tag{6}$$

Suppose the directed path from $K$ to $M$ is as follows:

$$K \to \theta_1 \to \dots \theta_{m''} \to M \tag{7}$$

Recall that $M$ is a non-collider along $q$. Thus the subpath of $q$ between $M$ and $Y$ must start with a tail as $M \to \dots Y$. Now observe that if the subpath of $q$ between $M$ and $Y$ had no collider, then we would have the following directed path from $X$ to $Y$:

$$X \to U_1 \to U_m \to K \to \theta_1 \to \dots \to \theta_{m''} \to M \to \dots Y \tag{8}$$

However, we know $X$ is a non-ancestor of $Y$. Thus, there must be at least one collider between $M$ and $Y$ along $p$, all of which are open given $T'$. Consider the collider that is closest to $M$. There is a directed path from $M$ to this collider, and a directed path from this collider to a member of $T'$ since it is open conditioned on $T'$. But this means there is a directed path from $K$ to a member of $T'$ since there is a directed path from $K$ to this collider. This is a contradiction since we obtained $T'$ by removing all descendants of $K$ from $T$.

This establishes the claim that removing descendants of the collider along any d-connecting path that starts with a tail at $X$ cannot introduce a d-connecting path that starts with an arrow into $X$ when $X \notin An(Y)$. □

**Lemma A.2.** *Consider a DAG $D$ where $X \notin An(Y)$ and $Y \notin An(X)$, $X, Y$ are non-adjacent and $k$-covered. Then conditioned on any subset $T : |T| \leq k$, there exists a d-connecting path between $X, Y$ that has an arrowhead at both $X$ and $Y$.*

*Proof.* For the sake of contradiction suppose, conditioned on $c$, there is no d-connecting path with an arrow into $X$ and an arrow into $Y$. Since neither $X$ is an ancestor of $Y$ nor $Y$ is an ancestor of $X$, it must be that all d-connecting paths have colliders on them. And all such colliders must be ancestors of $T$.

Consider such a path $p$ where the edge adjacent to $X$ has a tail at $X$. Let $K$ be the collider that is closest to $X$.

Thus we have

$$X \to U_1 \to \dots \to U_m \to K \leftarrow V \dots Y$$

for some $\{U_i\}_{i \in m}, V$. Since the path is open it must be that $K \in An(T)$. Let $T' = T - De(K)$, where $De(K)$ are all descendants of $K$. Clearly, $q$ is no longer open conditioned on $T'$. We investigate other open paths now, keeping in mind that $X, Y$ being $k$-covered implies that $X \not\perp\!\!\!\perp Y \,|\, T'$ since $|T'| \leq |T| \leq k$.

***Claim 1:*** *Removing the descendants of the collider closest to X from the conditioning set can only add d-connecting paths that start with a tail at X but no d-connecting paths that start with an arrowhead at X.*

*Proof of Claim 1:* Since $X$ is not an ancestor of $Y$, by Lemma A.1, we know that removing the descendants of $K$ from $T$ can only introduce d-connecting paths that are out of $X$. □

Now consider the d-connecting paths under conditioning on $T'$. We know that these paths must have a tail either at $X$ or at $Y$ since we started with such d-connecting paths by assumption and by Claim 1, removing $De(K)$ from $T$ can only introduce new d-connecting paths that have tails at $X$. Using the fact that no path that has an arrowhead into both endpoints are opened, we can use recursion and claim that we can make $X, Y$ d-separated by removing descendants of colliders (that are closest to the endpoint that is adjacent to a tail) of active paths, which gives the following:

***Claim 2:*** *There exists a set $T^*$ of size at most $k$ such that $X \perp\!\!\!\perp Y \,|\, T^*$, which leads to a contradiction since $X, Y$ are k-covered by assumption.*

*Proof of Claim 2:* Given claim 1, we can continue removing descendants of the colliders of the active paths that are closest to the tail-end node from the set $T$. Either no d-connecting path is left at some point in this process, or that we end up removing all the variables from the conditioning set. If the former, this is a contradiction since $X, Y$ are $k$-covered . If the latter, then this is another contradiction due to the following: This means that given empty set, paths that have a tail adjacent to one of the endpoints, i.e., the paths with colliders on them (since all paths that have a tail adjacent to one of the endpoints must have a collider because $X \notin An(Y)$ and $Y \notin An(X)$) are d-connecting, which is not possible. This proves Claim 2. □

Due to the symmetry between $X, Y$, the supposition that the only d-connecting paths must have a tail adjacent to either endpoint must be wrong, which proves the lemma. □

**Lemma A.3.** *Consider a DAG $D$ where $X \notin An(Y)$, $X, Y$ are non-adjacent and k-covered. Then conditioned on any subset $T : |T| \leq k$, there exists a d-connecting path between $X, Y$ that starts with an arrow into $X$.*

*Proof.* For the sake of contradiction, suppose otherwise. Given $T$, all the d-connecting paths start with a tail at $X$. We will show that we can find some $T'$ of size at most $k$ that d-separates $X, Y$, which lead to a contradiction since $X, Y$ are assumed to be $k$-covered .

Consider any path $q$ that is d-connecting given $T$ which starts with a tail at $X$. Since $X \notin An(Y)$, by Lemma A.1 it must be that this path has at least one collider on it. Let $K$ be the collider that is closest to $X$. Thus we have

$$X \to U_1 \to \ldots \to U_m \to K \leftarrow V \ldots Y$$

for some $\{U_i\}_i, V$. Since the path is open, this collider cannot be blocking the path $q$. It must be that $K \in An(T)$. Let $T' = T - De(K)$, where $De(K)$ are all descendants of $K$. Clearly, $q$ is no longer open conditioned on $T'$. We investigate other open paths now, keeping in mind that $X, Y$ being k-covered implies that $X \not\perp Y \,|T'$ since $|T'| \leq |T| \leq k$.

**Claim 1:** *Removing the descendants of the collider closest to $X$ from the conditioning set can only add d-connecting paths that start with a tail at $X$ but no d-connecting paths that start with an arrowhead at $X$.*

*Proof of Claim 1:* Since $X$ is not an ancestor of $Y$, by Lemma A.1 we know that removing the descendants of $K$ from $T$ can only introduce d-connecting paths that are out of $X$. □

Now consider the d-connecting paths under conditioning on $T'$. We know that these paths must have a tail at $X$ since we started with only such d-connecting paths by assumption and by Claim 1, removing $De(K)$ from $T$ can only introduce d-connecting paths that have tails at $X$. Using the fact that no path that has an arrowhead into $X$, we can use recursion and claim that we can make $X, Y$ d-separated by removing descendants of colliders (that are closest to $X$) of active paths, which gives the following:

**Claim 2:** *There exists a set $T^*$ of size at most $k$ such that $X \perp\!\!\!\perp Y \,|T^*$, which leads to a contradiction since $X, Y$ are k-covered by assumption.*

*Proof of Claim 2:* Given claim 1, we can continue removing descendants of the first colliders of active paths that are closest to $X$ from the set $T$. Either, no d-connecting path is left at some point in this process or that we end up removing all the variables from the conditioning set. If the former, this is a contradiction since $X, Y$ are k-covered. If the latter, then this is another contradiction due to the following: This means that given empty set, paths that start with a tail and have colliders on them (as they cannot be directed and collider-free since $X$ is not an ancestor of $Y$) are d-connecting, which is not possible. This proves Claim 2. □.

Therefore, the supposition that all the d-connecting paths must have a tail adjacent to $X$ must be wrong, which proves the lemma. □

The above lemmas will be crucial in proving Lemma 3.6. Now consider a d-connecting path between $x, z$ given $c$ and a d-connecting path between $z, y$ given $c$. We have the following lemma:

**Lemma A.4.** *Let $p$ be an active path between $x, z$ given $c$, and $q$ be an active path between $z, y$ given $c$, where $x, y, z \notin c$. If $x$ and $y$ are d-separated given $c$, then*

1. *Paths $p, q$ must have no overlapping nodes and*

2. *$Y$ must be a collider along the concatenated path and $Y \notin An(c)$.*

*Proof.* We would like to allow the possibility that these paths might go through the same nodes. To address this, it helps to consider walks.

**Definition A.5.** A walk on a DAG is any sequence of edges.

**Definition A.6.** A path on a DAG is a sequence of edges where each node appears at most once.

There is a direct relation between active walks and d-connecting paths [8]. Indeed, this relation is leveraged to efficiently check dependence using paths, rather than having to search over all walks, which is a much larger space.

**Definition A.7.** A walk between two nodes $a, b$ is called active given $c$ if each collider along the walk is in $c$ and each non-collider is not in $c$.

**Definition A.8.** A path between two nodes $a, b$ is called open given $c$ if each collider along the path is in $An(c)$ and each non-collider is not in $c$.

Consider an active walk where a node $t$ appears multiple times. Observe that $t$ must appear with the same collider status, since otherwise the walk would not be active. If the appearance is of the form

$$a \ldots \xrightarrow{\alpha} t \rightarrow \ldots \leftarrow t \xleftarrow{\beta} \ldots b, \tag{9}$$

then there must be a collider that is in $c$ between the two appearances of $t$'s. We can skip the intermediate subpath between the two appearances of $t$'s to obtain the walk

$$a \ldots \xrightarrow{\alpha} t \xleftarrow{\beta} \ldots b, \tag{10}$$

Since there is at least one collider that is in $c$ along the skipped subpath, we have that $t \in An(c)$. Therefore, $t$ will not be blocking the path that is obtained after repeatedly applying this and other shortening steps. If the appearances is of the form:

$$a \ldots \xrightarrow{\alpha} t \rightarrow \ldots \rightarrow t \xrightarrow{\beta} \ldots b, \tag{11}$$

we can similarly skip the subpath between the two appearances of $t$'s to obtain the shorter walk

$$a \ldots \xrightarrow{\alpha} t \xrightarrow{\beta} \ldots b, \tag{12}$$

and repeat this process until $t$ is not repeated. The resulting walk/path is still open since $t$ will appear in the same collider status, namely as a non-collider and if it was not blocking the walk, it will not be blocking the path either. This argument holds for any configuration where $t$ is a non-collider. If the appearances is of the form:

$$a \ldots \xrightarrow{\alpha} t \leftarrow \ldots \rightarrow t \xleftarrow{\beta} \ldots b, \tag{13}$$

then it must be that $t \in c$, and thus the walk obtained by skipping the subwalk between the two appearances of $t$'s, i.e.,

$$a \ldots \xrightarrow{\alpha} t \xleftarrow{\beta} \ldots b, \tag{14}$$

must be d-connecting.

This shows that each active walk corresponds to a d-connecting path and vice verse. Now we can proceed with the proof of the lemma:

*1.* Suppose $p, q$ have overlapping nodes. Let $w_p$ be the walk that corresponds to $p$ and $w_q$ be the walk that corresponds to $q$. Consider the concatenated walk $w = w_p, w_q$. If any repeated node has different collider status along $w$, then the path is not active. But this means that that node was blocking either $w_p$ or $w_q$, which would be a contradiction. Therefore, repeated nodes cannot have different collider status along $w$.

Suppose a node $t$ is repeated in $w_p$ and $w_q$ and has the same collider status. In this case, consider the walk obtained by concatenating the sub-walk of $w_p$ between $x$ and $t$, with the sub-walk of $w_q$ from $t$ to $y$. By repeating this process for any repeated node, we can obtain a path that corresponds to this walk, which would always be active since the repeated nodes have the same collider status along this path that they had in $w_p$ or $w_q$ and were not blocking these walks. Therefore, they cannot block the concatenated path obtained this way either, which is a contradiction since we are given that $x, y$ are d-separated given $c$. Therefore if any node is repeated in $w_p$ and $w_q$ then the concatenated walk is always active. Thus, it must be the case that there is no repeated nodes.

*2.* Since there is no repeated nodes from *1.*, we can operate at the path level instead of considering walks. Suppose $Y$ is not a collider. Since $Y \notin c$, it would be d-connecting and thus the concatenating path would be d-connecting, a contradiction. Suppose $Y$ is a collider but $Y \in An(c)$. In this case, $Y$ would not block the concatenated path either, which is a contradiction. This establishes the result. $\quad\square$

The next lemma shows that colliders that are closed in $D$ must remain closed in the $k$-closure $\mathcal{C}_k(D)$.

**Lemma A.9.** *If a collider is blocked in $D$ conditioned on some $c : |c| \leq k$ then it must also be blocked in $\mathcal{C}_k(D)$ conditioned on c.*

*Proof.* Suppose $(X \to Z \leftarrow Y)_D$ is a collider that is blocked given $c$. Thus, it must be that $Z \notin An(c)$ in $D$. For the sake of contradiction, suppose that this collider is unblocked in $\mathcal{C}_k(D)$. Thus, it must be the case that $Z \in An(c)$ in $\mathcal{C}_k(D)$. This means there is a new directed path from $Z$ to $c$ in $\mathcal{C}_k(D)$. If this path existed in $D$, the collider would be unblocked, which is a contradiction. Thus, at least one of the edges along this path must have been added during the construction of $\mathcal{C}_k(D)$. Consider the collection of edges on this path that do not exist in $D$. Note that by construction of $\mathcal{C}_k(D)$, a directed edge $\alpha \to \beta$ is added between a $k$-covered pair $\alpha, \beta$ only if there is a directed path from $\alpha$ to $\beta$. Consider the path obtained by replacing the directed edge between any $k-covered$ pair along this path with the corresponding directed path in $D$. The resulting directed path must be in $D$. This shows that there was at least one path already in $D$ that implied $Z \in An(c)$, which is a contradiction. Therefore, any collider in $p$ that is unblocked in $\mathcal{C}_k(D)$ must also be unblocked in $D$. $\qquad\square$

**Lemma A.10.** *The set of ancestors of any set $c$ of nodes in $\mathcal{C}_k(D)$ are identical to the set of ancestors of $c$ in $\mathcal{C}_k(D)$.*

*Proof.* Adding edges to a graph, directed or bidirected, cannot decrease the set of ancestors of any node. We only need to show that the set of ancestors in $\mathcal{C}_k(D)$ is not larger than the set of ancestors in $D$.

Suppose otherwise: A node $a \in An(c)$ in $\mathcal{C}_k(D)$ but $a \notin An(c)$ in $D$. This can only happen if a collection of edges added during the construction of $\mathcal{C}_k(D)$ render $a$ an ancestor of $c$. However, each such edge is added only if there is a directed path between its endpoints in $D$. Consider the path obtained by replacing each such added edge along the path that renders $a$ an ancestor of $c$ in $\mathcal{C}_k(D)$ with the corresponding directed paths in $D$. This directed path must be in $D$, which means that $a$ was an ancestor of $c$ in $D$ as well, which is a contradiction. $\qquad\square$

We are finally ready for the proof of Lemma 3.6.

### Proof of Lemma 3.6:

Since no edge is removed during the construction of the $k$-closure, one direction immediately follows: If $a \perp\!\!\!\perp b \mid c$ in $\mathcal{C}_k(D)$, then $a \perp\!\!\!\perp b \mid c$ in $D$. The implication is clearly true for any set $c$ of size at most $k$ as well. Therefore we only need to show the other direction.

Suppose $a \perp\!\!\!\perp b \mid c$ in $D$ where $|c| \leq k$. We will show that $a \perp\!\!\!\perp b \mid c$ in $\mathcal{C}_k(D)$. For the sake of contradiction, suppose otherwise. Then there must be a d-connecting path $p$ between $a, b$ given $c$ in $\mathcal{C}_k(D)$. The length of any such path must be greater than 1 since otherwise, whether this edge already existed in $D$ or it was added during the construction of $\mathcal{C}_k(D)$, $a, b$ must have been dependent given $c$ in $D$, which is a contradiction. Since the orientation of the existing edges in $D$ did not change in $\mathcal{C}_k(D)$, either this path did not exist in $D$ or that it existed but it was blocked by some collider that is not in $An(c)$ in $D$. The latter is not possible due to Lemma A.9, since any unblocked collider in $\mathcal{C}_k(D)$ must also be unblocked in $D$. Thus, it must be that this d-connecting path did not exist in $D$.

***Suppose $p$ does not exist in $D$.*** At least one edge must have been added to form this path in $\mathcal{C}_k(D)$ during the construction of $\mathcal{C}_k(D)$.

For any added edge $u \to v$, the following is true: Since $u \to v$ was added in $\mathcal{C}_k(D)$, it must be the case that $v \notin An(u)$ since otherwise there would be a cycle. By Lemma A.3, conditioned on $c$, there exists a d-connecting path between $u, v$ where the edge adjacent to $v$ is into $v$. For any added edge $u \leftrightarrow v$, the following is true: Since $u \leftrightarrow v$ was added in $\mathcal{C}_k(D)$, it must be the case that $u \notin An(v)$ and $v \notin An(u)$. By Lemma A.2, conditioned on $c$, there exists a d-connecting path between $u, v$ where the edge adjacent to $u$ is into $u$ and the edge adjacent to $v$ is into $v$. Call any such path implied by these lemmas a *replacement path*. Note that a replacement path might be directed or not.

Consider a path $q$ in $D$ that is obtained from $p$ by switching the edges added during the construction of $\mathcal{C}_k(D)$ with the replacement paths using the following policy: Suppose $u \to v$ in $\mathcal{C}_k(D)$ for some $k$-covered pair $u, v$. If a directed path is open given $c$ in $D$, use that path as the replacement path

for the edge $a \to b$. If not, use any other path. This means that either the path that replaces an edge $u \to v$ is directed or that both the endpoints have an arrowhead and that $u \in An(c)$.

Observe that each replacement path is d-connecting and the subpaths of $p$ that remain intact in $q$ must be d-connecting since $p$ is d-connecting. By Lemma A.4, any two paths – whether it is a pair of replacement paths or a replacement path and a subpath of $p$ – have overlapping nodes, then their concatenation must be d-connecting. Since we assumed that $q$ was not d-connecting, it must be that one of the endpoints of one of the added edges must be blocking $q$. We investigate each such node to verify that $q$ is indeed d-connecting to arrive at a contradiction.

In other words, the collider status of some of these nodes must have changed due to replacing some edges with replacement paths. Specifically due to Lemma A.4, one of the endpoints of replacement paths must be a collider and not an ancestor of $c$. Since ancestrality status cannot change from $D$ to $\mathcal{C}_k(D)$ due to Lemma A.10, the only way for $q$ to not be d-connecting is if some node that is not an ancestor of $c$ changes status from being a non-collider along $p$ to being a collider along $q$.

Note that for an edge $u \leftrightarrow v$, the nodes $u$ and $v$ are adjacent to an arrowhead in the replacement path. Thus, if some node $t$ changes collider status in $q$ compared to $p$, it cannot be due to bidirected edges along $p$.

Now consider the directed edges $u \to v$. If the replacement path is directed from $u$ to $v$, similarly $u$ is adjacent to a tail and $v$ is adjacent to an arrowhead on the replacement path. Therefore, such edges cannot alter the collider status of nodes at the junction of different paths. Finally consider the directed edges $u \to v$ where $u$ and $v$ are both adjacent to an arrowhead on the replacement path. Observe that this edge cannot change the collider status of $v$. We now focus on $u$. If the other edge adjacent to $u$ along $q$ is a tail, $u$ remains a non-collider and cannot block $q$. Now suppose the other edge adjacent to $u$ along $q$ is an arrowhead. This makes $u$ a collider in $q$ whereas $u$ was a non-collider along $p$ since we had $u \to v$ along $p$. However, by construction of $q$, as we ended up adding a path with arrowheads at both endpoints, it must be that the directed path between $u, v$ (which exists since $u \to v$ was added during the construction of $\mathcal{C}_k(D)$) must be blocked via conditioning. This means $u \in An(c)$ which means that although the status of $u$ changes from non-collider to collider, it must be that this collider does not block $q$ since it is an ancestor of $c$. Therefore, no replacement path can alter the status of a node to block the path $q$, and $q$ must be d-connecting, which contradicts with the assumption that the path was not d-connecting in $D$.

This establishes that any d-connecting path in $\mathcal{C}_k(D)$ is also d-connecting in $D$, which establishes that if $a \not\perp\!\!\!\perp b \,|\, c$ in $\mathcal{C}_k(D)$, then $a \not\perp\!\!\!\perp b \,|\, c$ in $D$. This establishes the lemma.

## A.2    Proof of Lemma 3.8

For a mixed graph to be a maximal ancestral graph, we need to show that it does not have directed or almost directed cycles and that any non-adjacent pair of nodes can be made conditionally independent by conditioning on some subset of observed variables [24]. We first define almost directed cycle, and propose a lemma that shows that $k$-closure graphs do not have directed or almost directed cycles.

**Definition A.11** ([24]).  A directed path $p$ from $a$ to $b$ and the edge $a \leftrightarrow b$ is called an almost directed cycle.

**Lemma A.12.** *For any DAG $D$, and integer $k$, $\mathcal{C}_k(D)$ does not have directed or almost directed cycles.*

*Proof.* Suppose, for the sake of contradiction that there is a directed cycle in $\mathcal{C}_k(D)$. Since each edge $X \to Y$ in $\mathcal{C}_k(D)$ either exists in $D$ or for each such edge in $\mathcal{C}_k(D)$, there is a directed path from $X$ to $Y$ in $D$, existence of a directed cycle in $\mathcal{C}_k(D)$ would imply a directed cycle in $D$, which contradicts with the DAG assumption of $D$.

Suppose, for the sake of contradiction that there is an almost directed cycle in $\mathcal{C}_k(D)$, i.e., we have a directed path from $a$ to $b$ for two nodes $a \leftrightarrow b$. Since $a \leftrightarrow b$ is added during construction of $\mathcal{C}_k(D)$, it must be the case that neither $a$ nor $b$ are ancestors of each other. However, from the above argument, there must be a directed path from $a$ to $b$ in $D$, which is a contradiction. Thus, $\mathcal{C}_k(D)$ cannot have almost directed cycles. $\square$

The other condition for a mixed graph to be a maximal ancestral graph is that for any non-adjacent pair of nodes, there exists a subset of the observed variables that make them conditionally independent. For the $k$-closure graphs, this simply follows by construction: Any pair of nodes that are non-adjacent in $\mathcal{C}_k(D)$ can be made conditionally independent given some set of size at most $k$ in $D$ by construction of $\mathcal{C}_k(D)$. From Lemma 3.6, this conditional independence relation must be retained in $\mathcal{C}_k(D)$. Thus any non-adjacent pair of nodes in $\mathcal{C}_k(D)$ can be d-separated in $\mathcal{C}_k(D)$ by some conditioning set of size at most $k$. This establishes the claim.

$\square$

## A.3 Proof of Theorem 3.9

Our main observation is that a parallel of Lemma A.2 works for MAGs with $k$-covered bidirected edges. The following lemmas are for any mixed graph $\mathbb{K}$ that satisfies the constraints in Theorem 3.9, i.e., those that are MAGs and that satisfy the condition that for any bidirected edge $a \leftrightarrow b$, $a, b$ are $k$-covered in the graph $\mathbb{K} - (a \leftrightarrow b)$.

**Lemma A.13.** *Suppose $X \notin An(Y)$. Suppose there is a d-connecting path $p$ between $X, Y$ given $T$ that starts with an arrow out of $X$.*

    *1. There is at least one collider along $p$.*

    *2. Let $K$ be the collider closest to $X$ on $p$. Then conditioning on $T' = T - De(K)$ instead of $T$ does not introduce new d-connecting paths that start with an arrowhead at $X$.*

*Proof.* 1. Any path that starts with $X \to \dots$ must either be directed, or there must be at least one collider along the path. Since the path is between $X, Y$ and $X \notin An(Y)$, it must be that the path has at least one collider on it.

2. First note that without loss of generality, $p$ has the following form for some integer $m \geq 0$ ($m = 0$ means $X \to K$):

$$X \to U_1 \to U_2 \dots \to U_m \to K \leftarrow* \dots Y. \tag{15}$$

$*$ is a wildcard representing either an arrowhead or a tail.

Suppose for the sake of contradiction that conditioned on $T'$, there is a new d-connecting path $q$ that starts with an arrowhead into $X$. $q$ was clearly closed conditioned on $T$ and became open by us removing nodes from the conditioning set $T$. This can only happen if we removed some node from $T$ that is a non-collider along $q$. Consider the non-collider we removed that was closest to $X$, call this $M$. Thus we have the path $q$ that looks like this:

$$X \leftarrow* W \dots M \dots Y, \tag{16}$$

where $M$ is a non-collider on this path and is in $De(K)$.

We observe that the subpath between $M$ and $X$ cannot be directed from $M$ to $X$. Because this would create the following cycle, since $K$ is assumed to be the first collider along $p$, and an ancestor of $M$.

$$M \to \dots \to X \to U_1 \dots U_m \to K \to \dots \to M. \tag{17}$$

Thus a closer look at the path $q$ reveals the following structure for some integer $m'$ and node $V$:

$$X \leftarrow W_1 \leftarrow W_2 \dots \leftarrow W_{m'} *\to V \dots M \dots Y \tag{18}$$

We will consider the following two cases: The edge mark adjacent to $M$ along the subpath between $W_{m'}$ and $M$ is a tail or an arrowhead.

**Suppose the edge mark adjacent to $M$ along the subpath between $W_{m'}$ and $M$ is a tail:** This means there is at least one collider between $W_{m'}$ and $M$. Any such collider must be active since $q$ is active given $T'$. Consider the collider that is closest to $M$. Since it is active, this collider must be an ancestor of $T'$. However, observe that $K$ is an ancestor of this collider which implies that $K$ is an ancestor of $T'$ as well. However, we obtained $T'$ by removing all descendants of $K$ from $T$, which is a contradiction.

**This establishes that the edge mark adjacent to $M$ along the subpath between $W_{m'}$ and $M$ is an arrowhead.** Thus, this reveals the following structure for $q$:

$$X \leftarrow W_1 \leftarrow W_2 \dots \leftarrow W_{m'} *\to \dots *\to M \dots Y \tag{19}$$

Suppose the directed path from $K$ to $M$ is as follows for some $\{\theta_i\}_i$ for some integer $m''$:

$$K \to \theta_1 \to \dots \theta_{m''} \to M \tag{20}$$

Recall that $M$ is a non-collider along $q$. Thus, the subpath of $q$ between $M$ and $Y$ must start with a tail as $M \to \dots Y$. Now observe that if the subpath of $p$ between $M$ and $Y$ had no collider, then we would have the following directed path from $X$ to $Y$:

$$X \to U_1 \to \dots \to U_m \to K \to \theta_1 \to \dots \to \theta_{m''} \to M \to \dots \to Y \tag{21}$$

However, we know $X$ is a non-ancestor of $Y$. Thus, there must be at least one collider between $M$ and $Y$ along $p$, all of which are open given $T'$. Consider the collider that is closest to $M$. There is a directed path from $M$ to this collider, and a directed path from this collider to a member of $T'$. But this means there is a directed path from $K$ to a member of $T'$ since there is a directed path from $K$ to this collider. This is a contradiction since we obtained $T'$ by removing all descendants of $K$ from $T$.

This establishes the claim that removing descendants of the collider along any d-connecting path that starts with a tail at $X$ cannot introduce a d-connecting path that starts with an arrow into $X$ when $X \notin An(Y)$. □

The following is the parallel lemma to Lemma A.2 for any mixed graph $\mathbb{K}$ that satisfies the conditions of Theorem 3.9.

**Lemma A.14.** *Consider a bidirected edge $X \leftrightarrow Y$ in $\mathbb{K}$. Suppose conditioned on any subset $T : |T| \leq k$, $X \not\perp\!\!\!\perp Y | T$ in $G - (X \leftrightarrow Y)$. Then conditioned on any $T : |T| \leq k$, there exists a d-connecting path between $X, Y$ that starts with an arrow into $X$ and an arrow into $Y$.*

*Proof.* For the sake of contradiction suppose, conditioned on some $T : |T| \leq k$, there is no d-connecting path with an arrow into $X$ and an arrow into $Y$. Since neither $X$ is an ancestor of $Y$ nor $Y$ is an ancestor of $X$, all d-connecting paths must have colliders on them. And all such colliders must be ancestors of $T$.

Consider such a path $p$ where, without loss of generality, the edge adjacent to $X$ has a tail at $X$. Let $K$ be the collider that is closest to $X$.

Thus we have
$$X \to U_1 \to \ldots \to U_m \to K \leftarrow\!* V \ldots Y \tag{22}$$
for some $\{U_i\}_i, V$ and integer $m$. Since the path is open, this collider must be unblocked. It must be that $K \in An(T)$. Let $T' = T - De(K)$, where $De(K)$ are all descendants of $K$. Clearly, $p$ is no longer open. We investigate other open paths now, keeping in mind that $X, Y$ are dependent given $T'$ since $|T'| \leq k$.

*Claim 1:* Removing the descendants of the collider closest to $X$ from the conditioning set can only add d-connecting paths that start with a tail at $X$ but no d-connecting path that starts with an arrowhead at $X$.

*Proof of Claim 1:* Since a bidirected edge exists between $X, Y$, and that $\mathbb{K}$ is a MAG, neither $X$ nor $Y$ are ancestors of one another, since then we would have an almost directed cycle. By Lemma A.13, we know that removing the descendants of $K$ from $T$ can only introduce d-connecting paths that are out of $X$. □

Now consider the d-connecting paths under conditioning on $T'$. We know that these paths must have a tail either at $X$ or at $Y$. Using the above claim that no path that has an arrowhead into both endpoints are opened, we can use recursion and claim that we can make $X, Y$ d-separated by removing descendants of colliders (that are closest to the endpoint that is adjacent to a tail) of active paths, which gives the following:

*Claim 2:* There exists a set $T^*$ of size at most $k$ such that $X \perp\!\!\!\perp Y | T^*$, which leads to a contradiction since $X, Y$ cannot be made independent by conditioning on sets of size at most $k$ by the assumption.

*Proof of Claim 2.* Given claim 1, we can continue removing descendants of the colliders of the active paths that are closest to the tail-end node from the set $T$. Either no d-connecting path is left at some point in this process, or that we end up removing all the variables from the conditioning set. If former, this is a contradiction since $X, Y$ cannot be made conditionally independent given empty set. If the latter is true, then there is another contradiction due to the following: This means that given empty set, paths that have a tail adjacent to one of the endpoints, i.e., the paths with colliders on them (since all paths that have a tail adjacent to one of the endpoints must have a collider because $X \notin An(Y)$ and $Y \notin An(X)$) are d-connecting, which is not possible. This proves Claim 2. □

Therefore, the supposition that the only d-connecting paths must have a tail adjacent to either endpoint must be wrong, which proves the lemma. □

*Proof of Theorem 3.9.* Now, we are ready to prove the main characterization theorem. We will need the following lemma:

**Lemma A.15.** *Let $\mathbb{K}$ be a mixed graph that satisfies the conditions in Theorem 3.9. Let $\mathbb{K}'$ be the graph obtained by removing all the bidirected edges from $\mathbb{K}$. Then*

1. *$\mathbb{K}'$ is a DAG and*

2. *$\mathbb{K}' \sim_k \mathbb{K}$.*

*Proof.* Since the only difference between $\mathbb{K}'$ and $\mathbb{K}$ is the removal of bidirected edges, any directed cycle that exists in $\mathbb{K}'$ would also have existed in $\mathbb{K}$, which contradicts with the assumption that $\mathbb{K}$ is a MAG. This establishes that $\mathbb{K}$ has no directed cycles.

Clearly, any independence statement in $\mathbb{K}$ holds in $\mathbb{K}'$, since it is obtained from $\mathbb{K}$ by removing edges. Thus any degree-$k$ d-separation relation that holds in $\mathbb{K}$ also holds in $\mathbb{K}'$. Therefore, we only need to show that for any $c$ of size at most $k$ $(a \not\perp\!\!\!\perp b\,|\,c)_{\mathbb{K}}$ implies $(a \not\perp\!\!\!\perp b\,|\,c)_{\mathbb{K}'}$.

Suppose for the sake of contradiction that $a \not\perp\!\!\!\perp b\,|\,c$ in $\mathbb{K}$ but $a \perp\!\!\!\perp b\,|\,c$ in $\mathbb{K}'$. Let $p$ be a d-connecting path between $a, b$ given $c$ in $\mathbb{K}$. This path must be closed in $\mathbb{K}'$. Since the only difference between the two graphs is the removal of bidirected edges, ancestrality relations cannot be different. Thus, it cannot be the case that a collider that was open in $\mathbb{K}$ is now closed in $\mathbb{K}'$ and is closing the path $p$. Any collider that was open must still be open. Thus, the only way for $p$ to be closed in $\mathbb{K}'$ is if some bidirected edge $X \leftrightarrow Y$ along $p$ is removed. However, by Lemma A.14, for any such bidirected edge in $\mathbb{K}$, and for any conditioning set of size at most $c$, we have a d-connecting path called a *replacement path* with an incoming edge to both $X$ and $Y$. Consider the path $q$ obtained by replacing every bidirected edge along $p$ with a corresponding replacement path. Since $a, b$ are d-separated by assumption, this path cannot be open. As this path is a concatenation of several d-connecting paths – either sub-paths of $p$, which must be open, or replacement paths which must be open, by Lemma A.4, they must have no overlapping nodes, and some node at the junction of these paths must be a collider and non-ancestor of $c$. However, since we replaced bidirected edges $X \leftrightarrow Y$ with paths of the form $X \leftarrow * \ldots * \rightarrow Y$, both $X$ and $Y$ must have the same collider status on both $p$ and $q$. Thus, they cannot be blocking $q$ since they are not blocking $p$. This means that $q$ is d-connecting in $\mathbb{K}'$, which is a contradiction. This proves the lemma that $\mathbb{K}$ and $\mathbb{K}'$ must entail the same degree-$k$ d-separation relations, which implies they are $k$-Markov equivalent. $\qquad\square$

The only if direction: Suppose a mixed graph is a $k$-closure graph, i.e., $\mathbb{K} = \mathcal{C}_k(D)$ for some DAG $D$ and has the edge $a \leftrightarrow b$. Suppose for the sake of contradiction that $a, b$ are not $k$-covered in $\mathbb{K} - (a \leftrightarrow b)$. Let $\mathbb{K}'$ be the graph obtained from $\mathbb{K}$ by removing all the bidirected edges. Note that $\mathbb{K}'$ is a DAG since $\mathbb{K}$ has no directed cycles. Also note that all edges in $D$ must appear in $\mathbb{K}'$ by construction of $k$-closure graphs. $D$ can therefore be obtained from $\mathbb{K}$ by removing edges. Thus, any d-separation statement in $\mathbb{K}$ must also hold in $D$. Therefore, $a, b$ must be conditionally independent given some subset $c$ of size at most $k$ in $D$. This means $\mathbb{K}$, in which $a, b$ are adjacent, cannot be the $k$-closure graph of $D$, which is a contradiction.

If direction: Suppose a mixed graph $\mathbb{K}$ satisfies the conditions in Theorem 3.9. By Lemma A.15, for any such mixed graph $\mathbb{K}$, there is a DAG whose $k$-closure is $\mathbb{K}$, which shows that any such $\mathbb{K}$ is a valid $k$-closure graph, proving the theorem. $\qquad\square$

## A.4 Proof of Lemma 3.12

Let $K_1 = \mathcal{C}_k(D_1), K_2 = \mathcal{C}_k(D_2)$ be two $k$-closure graphs with the same skeleton and unshielded colliders. Suppose for the sake of contradiction that there is a path $p$ that is discriminating for a triple $\langle u, Y, v \rangle$ in both such that $Y$ is a collider along $p$ in $\mathcal{C}_k(D_1)$ and a non-collider in $\mathcal{C}_k(D_2)$. Thus, in $\mathcal{C}_k(D_1)$ we have the path $p$ as

$$a* \to z_1 \leftrightarrow z_2 \leftrightarrow \ldots \leftrightarrow z_m \leftrightarrow u \leftrightarrow Y \leftrightarrow v \tag{23}$$

where $z_i \to v, \forall i$ and $u \to v$ and $a, v$ are non-adjacent. Note that we cannot have $Y \leftarrow v$ instead of $Y \leftrightarrow v$ since this would create the almost directed cycle $u \to v \to Y \leftrightarrow u$. The same path with $Y$ as a non-collider can take two configurations in $\mathcal{C}_k(D_2)$, either as

$$a* \to z_1 \leftrightarrow z_2 \leftrightarrow \ldots \leftrightarrow z_m \leftrightarrow u \leftrightarrow Y \to v \tag{24}$$

or as

$$a* \to z_1 \leftrightarrow z_2 \leftrightarrow \ldots \leftrightarrow z_m \leftrightarrow u \leftarrow Y \to v \tag{25}$$

Other paths where $Y$ is a non-collider would either render $u$ a non-collider, which cannot happen by definition of a discriminating path, or create a directed or almost directed cycle. Since $a, v$ are non-adjacent by definition of a discriminating path, there must be some $S : |S| \leq k$ where $(a \perp\!\!\!\perp v | S)_{\mathcal{C}_k(D_1)}$. Note that $S$ must include all $z_i$'s and $u$, and not include $Y$ since otherwise there would be d-connecting paths between $a, v$ in $\mathcal{C}_k(D_1)$ due to the discriminating path. This means that $(a \not\perp\!\!\!\perp v | S)_{\mathcal{C}_k(D_2)}$.

Since $u \leftrightarrow Y$ in $\mathcal{C}_k(D_1)$, by Lemma A.2, there must be a d-connecting path between $u, Y$ in $D_1$ conditioned on $S$ that has an arrowhead at $Y$. By construction, this path must also appear in $\mathcal{C}_k(D_1)$. Since the path is inherited from $D_1$, it does not have bidirected edges. Consider the shortest of all such d-connecting paths, call this path $q$. Let $X$ be the node adjacent to $Y$ along $q$. Thus, $q$ has the form

$$u \leftarrow \ldots \to X \to Y. \tag{26}$$

We have that $X \to Y$ in both $D_1$ and $\mathcal{C}_k(D_1)$. In $\mathcal{C}_k(D_1)$, we have $X \to Y \leftrightarrow v$. Since the edge between $Y, v$ has a tail at $Y$ in $\mathcal{C}_k(D_2)$, this collider cannot exist in $\mathcal{C}_k(D_2)$. Thus, it must be the case that this collider is shielded in $\mathcal{C}_k(D_1)$, i.e., $X$ and $v$ are adjacent in $\mathcal{C}_k(D_1)$. Since $\mathcal{C}_k(D_1), \mathcal{C}_k(D_2)$ have the same skeleton, they must also be adjacent in $\mathcal{C}_k(D_2)$.

Now consider the path obtained by concatenating the subpath of $p$ $a* \to \ldots u$, and the subpath of $q$ between $u$ and $X$, and the edge between $X$ and $v$ in $\mathcal{C}_k(D_1)$. Call this path $r$. Note that the subpath of $q$ is d-connecting given $S$, as well as the subpath of $p$ since $z_i$'s and $u$ are in $S$. Thus, unless $X$ is a collider on it, the path $r$ between $a, v$ will be open, which would lead to a contradiction since $a, v$ are d-separated given $S$ in $\mathcal{C}_k(D_1)$. Thus, the edge between $X, v$ must have an arrowhead at $X$. Let $W$ be the node before $X$ along $q$. Thus we have $W \to X \leftrightarrow v$ in $\mathcal{C}_k(D_1)$. Note that $X \leftarrow v$ is not possible since this would create an almost directed cycle $X \to Y \leftrightarrow v \to X$ in $\mathcal{C}_k(D_1)$.

Suppose this collider is unshielded and appears in $\mathcal{C}_k(D_2)$ as well: $W* \to X \leftarrow* v$ in $\mathcal{C}_k(D_2)$. Thus in $\mathcal{C}_k(D_2)$, we have $Y \to v* \to X \leftarrow* W$. Since $X, Y$ are adjacent, it must be that $X \leftarrow Y$ or $X \leftrightarrow Y$ to avoid a directed or almost directed cycle. Thus in $\mathcal{C}_k(D_2)$, we have $X \leftarrow* Y$. However, this creates the collider $W* \to X \leftarrow* Y$ in $\mathcal{C}_k(D_2)$. Note that this collider cannot appear in $\mathcal{C}_k(D_1)$ since the edge between $X, Y$ has a tail at $X$ in $\mathcal{C}_k(D_1)$. Thus the collider must be shielded, meaning that $W, Y$ must be adjacent, and both in $\mathcal{C}_k(D_2)$ and in $\mathcal{C}_k(D_1)$. Since we have $W \to X \to Y$ in $\mathcal{C}_k(D_1)$, the edge must be $W \to Y$ in $\mathcal{C}_k(D_1)$. Furthermore, similar to $X$, $W$ cannot be in the conditioning set since this would block the path $q$. This means there is a d-connecting path that has an arrowhead at $Y$ that is shorter than $q$, which is a contradiction.

Thus the collider $W \to X \leftrightarrow v$ in $\mathcal{C}_k(D_1)$ must be shielded. Similar to the above argument, $W$ must be a collider along the path constructed by concatenating the subpath $a* \to \ldots u$ of $p$, and the subpath of $q$ between $u$ and $W$, and the edge between $W$ and $v$ since otherwise this path would be open, which would contradict with $a \perp\!\!\!\perp v | S$. Let $V$ be the node next to $W$ along $q$. Thus we have $V \to W \to X$ along $q$ and $V \to W \leftarrow* v$ is a collider in $\mathcal{C}_k(D_1)$. In fact, it must be that $W \leftrightarrow v$ since otherwise there would be an almost directed cycle $v \to W \to X \to Y \leftrightarrow v$ in $\mathcal{C}_k(D_1)$.

Suppose the collider $V \to W \leftrightarrow v$ in $\mathcal{C}_k(D_1)$ is unshielded and also appears in $\mathcal{C}_k(D_2)$. Note that if $V, X$ were adjacent in $\mathcal{C}_k(D_1)$, the orientation would have to be as $V* \to X$ since otherwise there

would be a directed cycle $V \to W \to X \to V$ in $\mathcal{C}_k(D_1)$. But this would imply that there is a shorter path than $q$ that connects $u, Y$ and has an arrow into $Y$. Thus, $V, X$ must be non-adjacent in $\mathcal{C}_k(D_1)$ and hence in $\mathcal{C}_k(D_2)$. Thus, $\langle V, W, X \rangle$ is an unshielded non-collider in $\mathcal{C}_k(D_1)$ and must also be in $\mathcal{C}_k(D_2)$. Thus it must be that $W \to X$ in $\mathcal{C}_k(D_2)$. Since $v {*}{\to} W \to X$ in $\mathcal{C}_k(D_2)$, it must be that $v {*}{\to} X$ in $\mathcal{C}_k(D_2)$ to avoid a directed or almost directed cycle. Since $Y \to v {*}{\to} X$ in $\mathcal{C}_k(D_2)$, it must be that $X \leftarrow{*} Y$ to avoid a cycle or almost directed cycle in $\mathcal{C}_k(D_2)$. However, now we have a collider $W \to X \leftarrow{*} Y$ in $\mathcal{C}_k(D_2)$ that is a non-collider in $\mathcal{C}_k(D_1)$ since in $\mathcal{C}_k(D_1)$ we have $X \to Y$. Thus, this collider must be shielded, i.e, $W, Y$ must be adjacent in $\mathcal{C}_k(D_2)$. Thus, they must also be adjacent in $\mathcal{C}_k(D_1)$. Since $W \to X \to Y$ in $\mathcal{C}_k(D_1)$, it must be that $W \to Y$ in $\mathcal{C}_k(D_1)$ to avoid a cycle. But this means there is a shorter d-connecting path between $u, Y$ given $S$ with an arrowhead at $Y$, which is a contradiction.

Therefore, the collider $V \to W \leftrightarrow v$ must be shielded in $\mathcal{C}_k(D_1)$. We can repeat the above argument as many times as needed continuing from the parent of $V$ along $q$. As we keep shielding more and more colliders in $\mathcal{C}_k(D_1)$, eventually when we shield the first node along $q$ next to $u$, we will end up with a directed path from $u$ to $Y$. However, this is a contradiction since bidirected edge was added between $u, Y$ which implies that $u$ is not an ancestor of $Y$.

Therefore, if two $k$-closure graphs $\mathcal{C}_k(D_1), \mathcal{C}_k(D_2)$ have the same skeleton and unshielded colliders, then they cannot have different colliders along discriminating paths, which proves the lemma.

## A.5 Proof of Corollary 3.13

($\Rightarrow$) If they are Markov equivalent then by Lemma 3.8 they are two Markov equivalent MAGs. Therefore by Theorem 3.11 they have the same skeleton, and the same unshielded colliders.

($\Leftarrow$) If they have the same skeleton and the same unshielded colliders, then by Lemma 3.12, they must have the same colliders along discriminating paths. Thus, by Theorem 3.11 they are equivalent. $\square$

## A.6 Proof of Theorem 3.14

($\Rightarrow$) Suppose $D_1, D_2$ are $k$-Markov equivalent. For the sake of contradiction suppose that $\mathcal{C}_k(D_1)$ and $\mathcal{C}_k(D_2)$ are not Markov equivalent. By Corollary 3.13, this happens when either they have different skeletons, or different unshielded colliders. Thus, there are two cases:

***k-closures have different skeletons:*** $\mathcal{C}_k(D_1)$ and $\mathcal{C}_k(D_2)$ have different skeletons. Suppose without loss of generality that $\mathcal{C}_k(D_1)$ has an extra edge, i.e., $a, b$ are adjacent in $\mathcal{C}_k(D_1)$ but not in $\mathcal{C}_k(D_2)$. This can only happen if $\exists S \subset V : |S| \leq k$ such that $(a \perp\!\!\!\perp b \,|S)_{D_2}$, while there is no such separating set in $D_1$, implying that $(a \not\perp\!\!\!\perp b \,|S)_{D_1}$. This is a contradiction with the supposition that $D_1, D_2$ are k-Markov equivalent. Therefore, $\mathcal{C}_k(D_1)$ and $\mathcal{C}_k(D_2)$ must have the same skeletons.

For completeness, we restate the definition of unshielded collider in $k$-closure graphs, which is identical to how it is defined in MAGs.

**Definition A.16.** A triple $\langle a, c, b \rangle$ in a $k$-closure graph is called an unshielded collider if $a, b$ are non-adjacent, $a, c$ and $c, b$ are adjacent and the edges adjacent to $c$ have an arrowhead mark at $c$.

According to the definition, a triple $\langle a, c, b \rangle$ in a $k$-closure graph $\mathcal{C}_k(D)$ can be an unshielded collider if the induced subgraph on the nodes take either of the following configurations:

1. $a \rightarrow c \leftarrow b$
2. $a \rightarrow c \leftrightarrow b$
3. $a \leftrightarrow c \leftarrow b$
4. $a \leftrightarrow c \leftrightarrow b$

We use asterisk to represent either an arrowhead or a tail. $\ast\!\rightarrow$ represents either $\rightarrow, \leftrightarrow$. Similarly, $\leftarrow\!\ast$ represents either $\leftarrow$ or $\leftrightarrow$.

***k-closures have different unshielded colliders:*** Without loss of generality assume that $(a\ast\!\rightarrow c \leftarrow\!\ast b)_{\mathcal{C}_k(D_1)}$ but this unshielded collider does not exist in $\mathcal{C}_k(D_2)$, i.e., $\langle a, c, b \rangle$ is an unshielded non-collider in $\mathcal{C}_k(D_2)$.

**Lemma A.17.** *In a $k$-closure graph $\mathcal{C}_k(D)$, two nodes $a, b$ are non-adjacent iff $(a \perp\!\!\!\perp b \,|S)_{\mathcal{C}_k(D)}$ for some $S \subset V$.*

*Proof.* ($\Rightarrow$) Suppose $a, b$ are non-adjacent in $\mathcal{C}_k(D)$. Thus it must be the case that $(a \perp\!\!\!\perp b \,|S)_D$ for some $S$ such that $|S| \leq k$, since otherwise $a, b$ would be made adjacent during the construction of $\mathcal{C}_k(D)$. By Lemma 3.6, this means $(a \perp\!\!\!\perp b \,|S)_{\mathcal{C}_k(D)}$. This establishes the only if direction.

($\Leftarrow$) Suppose now that $(a \perp\!\!\!\perp b \,|S)_{\mathcal{C}_k(D)}$. By definition of d-separation, adjacent nodes cannot be d-separated and thus $a, b$ must be non-adjacent in $\mathcal{C}_k(D)$. $\qquad\square$

**Lemma A.18.** *In a $k$-closure graph $\mathcal{C}_k(D)$, any pair of non-adjacent nodes $a, b$ are separable by a set of size at most $k$, i.e., $\exists S : |S| \leq k, (a \perp\!\!\!\perp b \,|S)_{\mathcal{C}_k(D)}$.*

*Proof.* Suppose otherwise: For some non-adjacent pair $a, b$ all d-separating sets in $\mathcal{C}_k(D)$ have size greater than $k$. Let $S$ be the smallest subset that makes $a, b$ d-separated, i.e., $(a \perp\!\!\!\perp b \,|S)_{\mathcal{C}_k(D)}$ – one exists by Lemma A.17. Clearly, $|S| > k$. Note that non-adjacency of $a, b$ in $\mathcal{C}_k(D)$ implies that $a, b$ are separable in $D$ with some set $T$ of size at most $k$: $(a \perp\!\!\!\perp b \,|T)_D$. Since $|T| \leq k < |S|$ and $S$ is the smallest subset that d-separates $a, b$ in $\mathcal{C}_k(D)$, it must be that $(a \not\perp\!\!\!\perp b \,|T)_{\mathcal{C}_k(D)}$. However, this contradicts with Lemma 3.6 which says that $D$ and $\mathcal{C}_k(D)$ must entail the same d-separation constraints for conditioning sets of size up to $k$. $\qquad\square$

Since $a, b$ are non-adjacent in both graphs, by Lemma A.18 there are two subsets $S_1, S_2$ of size at most $k$ such that

$$(a \perp\!\!\!\perp b \,|S_1)_{\mathcal{C}_k(D_1)}, (a \perp\!\!\!\perp b \,|S_2)_{\mathcal{C}_k(D_2)}. \tag{27}$$

Clearly, $S_1 \not\ni c, S_2 \ni c$ since $c$ is a collider between $a, b$ in $\mathcal{C}_k(D_1)$ and a non-collider in $\mathcal{C}_k(D_2)$. If we switch the conditioning sets, due to the different collider status of $c$ in both graphs the d-separation statements will switch to d-connection statements:

$$(a \not\perp\!\!\!\perp b \,|S_1)_{\mathcal{C}_k(D_2)}, (a \not\perp\!\!\!\perp b \,|S_2)_{\mathcal{C}_k(D_1)}. \tag{28}$$

Since $S_1$ and $S_2$ have size of at most $k$, then from Lemma 3.6 we have that:

$$(a \perp\!\!\!\perp b \,|\, S_1)_{D_1}, (a \not\perp\!\!\!\perp b \,|\, S_1)_{D_2} \tag{29}$$

This implies that $D_1, D_2$ are not $k$-Markov equivalent which is a contradiction.

This establishes that if $D_1, D_2$ are $k$-Markov equivalent then $\mathcal{C}_k(D_1), \mathcal{C}_k(D_2)$ must have the same skeleton and the same unshielded colliders. By Corollary 3.13, $\mathcal{C}_k(D_1), \mathcal{C}_k(D_2)$ are Markov equivalent.

($\Leftarrow$)

Suppose that $\mathcal{C}_k(D_1)$ and $\mathcal{C}_k(D_2)$ are Markov equivalent. Then they impose the same d-separation statements. Therefore they impose the same d-separation statements when the conditioning set is restricted to size at most $k$. By Lemma 3.6, this means that $D_1, D_2$ must also impose the same d-separation statements for conditioning sets of size of at most $k$. This establishes that $D_1, D_2$ are $k$-Markov equivalent. $\qquad\square$

## A.7 Proof of Lemma 4.1

Suppose in the $k$-closure graph $\mathcal{C}_k(D)$ for some DAG $D$ and integer $k$, we have a discriminating path for $Y$ between the nodes $a, v$ of the form

$$a* \to \leftrightarrow \ldots \leftrightarrow u \leftrightarrow Y \leftrightarrow v.$$

By definition of discriminating path, $u$ must be a collider along the path, and $u \to v$. If $Y \leftarrow v$, then we would have an almost directed cycle $u \to v \to Y \leftrightarrow u$. Thus, we have $u \leftrightarrow Y \leftrightarrow v$.

First, we show that the arrowhead at $Y$ of the edge $Y \leftrightarrow v$ can be learned by first orienting unshielded colliders and then applying $\mathcal{R}1$ and $\mathcal{R}2$. Consider the bidirected edge $u \leftrightarrow Y$. By definition of discriminating path $a, v$ must be non-adjacent and thus separable by a set of size at most $k$ by Lemma A.18. Therefore, we have a set $S : |S| \leq k$ such that $a \perp\!\!\!\perp v \,|\, S$. By the discriminating path definition, every collider along the path must be a parent of $v$, and therefore it must be the case that every collider along the discriminating path including $u$ must be in $S$, since otherwise there would be a d-connecting path. From Lemma A.2, conditioned on any set of size at most $k$, we have a d-connecting path that starts with an edge into $Y$. Consider the shortest such path $q$. Let $X$ be the node immediately before $Y$ along $q$. Since this path exists in the DAG by the lemma, we have $X \to Y$. If $X, v$ are non-adjacent, then $X \to Y \leftrightarrow v$ would be an unshielded collider and we are done.

Suppose $X, v$ are adjacent. Note that conditioned on $S$, $a$ and $X$ are d-connected. If $X$ is a non-collider along the path obtained by concatenating the subpath of $q$ between $a, X$ and the edge between $X, v$, then $a, v$ would be d-connected given $S$, which is a contradiction. Therefore, $X$ must be a collider along this path. Thus we have $X \leftrightarrow v$. Note that we cannot have $X \leftarrow v$ since this would create an almost directed cycle in $\mathcal{C}_k(D)$. Let $V$ be the node immediately before $X$ along $q$. Thus we have $V \to X \leftrightarrow v$ (not $V \leftrightarrow X$ since this edge exists in $D$).

Suppose $V, v$ are non-adjacent. Thus, the collider $V \to X \leftarrow* v$ is unshielded, and therefore can be learned. Furthermore, $V, Y$ must be non-adjacent since otherwise, we must have $V \to Y$ to avoid a cycle and there would be a path that is shorter than $q$, which "jumps over" the node $X$ along $q$. Thus, we can learn that $X \to Y$ from $\mathcal{R}1$. Finally, since now we have learned $v* \to X \to Y$ and that $Y, v$ are adjacent, by $\mathcal{R}2$, we must have $Y \leftarrow* v$. Thus the arrowhead mark at $Y$ of the edge $Y \leftrightarrow v$ can be learned, and we are done.

Suppose $V, v$ are adjacent. Following a similar argument, we either have some unshielded collider that can be propagated using the argument above to orient $Y \leftarrow* v$, or we can continue covering unshielded colliders, which would imply the previous nodes are always parents along $q$. But this implies that $u$ has a directed path to $Y$, which cannot happen since we have $u \leftrightarrow Y$ in $\mathcal{C}_k(D)$. This establishes that $Y \leftarrow* v$ can be learned by orienting unshielded colliders and applying the rules $\mathcal{R}1$ and $\mathcal{R}2$.

For the arrowhead mark at $Y$ of the edge $u \leftrightarrow Y$, similarly consider the shortest d-connecting path $q$ between $Y, v$ in $D$ given $S$ that starts with an arrow into $Y$. The argument follows similarly that either there would be directed path from $v$ to $Y$, which is a contradiction with the existence of the edge $Y \leftrightarrow v$ in the $k$-closure graph, or that there exists an unshielded collider along $q$ that can be learned by orienting unshielded colliders, which can be propagated to learn $u* \to Y$ using rules $\mathcal{R}1$, and $\mathcal{R}2$. This establishes the lemma. $\qquad\square$

## A.8 Proof of Theorem 4.4

We show soundness of the two new rules with the following two lemmas:

**Lemma A.19.** *Let $K$ be a mixed graph that is sandwiched between $\varepsilon_k(D)$ and PAG($\mathcal{C}_k(D)$), i.e., $\varepsilon_k(D) \subseteq K \subseteq PAG(\mathcal{C}_k(D))$. $\mathcal{R}11$ is sound on $K$ for learning the k-essential graph, i.e., if $K' = \mathcal{R}11(K)$, then $\varepsilon_k(D) \subseteq K' \subseteq K$.*

*Proof.* For the sake of contradiction, suppose otherwise: $\mathcal{R}11$ orients an edge $ao\rightarrow b$ in $K$ as $a \rightarrow b$, and there is a DAG $D'$ with a $k$-closure graph $\mathcal{C}_k(D')$ that is Markov equivalent to $\mathcal{C}_k(D)$ and is consistent with $K$ where $a \leftrightarrow b$. This means $a, b$ are $k$-covered in $D'$. Then from Lemma A.2, conditioned on any subset $S$ of size at most $k$, there must be a d-connecting path that starts with an arrow into both $a$ and $b$ in $D$. By construction of the $k$-closure graph, this path must also exist in $\mathcal{C}_k(D')$. Therefore, there must be some node $w$ such that $a \leftarrow w$. Since $a$ has no incoming edges, it must be the case that $w \in C$. However, $b$ is chosen so that $b$ is non-adjacent to any node in $C$. Therefore, $b$ must be non-adjacent to $w$ in $K$. However, this creates the unshielded collider $w \rightarrow a \leftrightarrow b$. However, note that $wo\text{---}oao\rightarrow b$ in PAG($\mathcal{C}_k(D)$), and thus $\langle w, a, b \rangle$ is a non-collider in $\mathcal{C}_k(D)$. Therefore, $\mathcal{C}_k(D')$ cannot be Markov equivalent to $\mathcal{C}_k(D)$, which is a contradiction. □

**Lemma A.20.** *Let $K$ be a mixed graph that is sandwiched between $\varepsilon_k(D)$ and PAG($\mathcal{C}_k(D)$), i.e., $\varepsilon_k(D) \subseteq K \subseteq PAG(\mathcal{C}_k(D))$. $\mathcal{R}12$ is sound on $K$ for learning the k-essential graph, i.e., if $K' = \mathcal{R}12(K)$, then $\varepsilon_k(D) \subseteq K' \subseteq K$.*

*Proof.* For the sake of contradiction, suppose otherwise: $\mathcal{R}12$ orients an edge $ao\text{---}oc$ in $K$ as $a\text{---}c$, and there is a DAG $D'$ with a $k$-closure graph $\mathcal{C}_k(D')$ that is Markov equivalent to $\mathcal{C}_k(D)$ and is consistent with $K$ where $a \leftrightarrow c$. This means $a, c$ are $k$-covered in $D'$. Then from Lemma A.2, conditioned on any subset $S$ of size at most $k$, there must be a d-connecting path that starts with an arrow into both $a$ and $c$ in $D$. By construction of the $k$-closure graph, this path must also exist in $\mathcal{C}_k(D')$. Therefore, there must be some node $w$ such that $a \leftarrow w$. Since $a$ has no incoming edges, it must be the case that $w \in C$. However, $c$ is chosen so that $c$ is non-adjacent to any other node in $C$. Therefore, $c$ must be non-adjacent to $w$ in $K$. However, note that $wo\text{---}oao\text{---}oc$ in PAG($\mathcal{C}_k(D)$), and thus $\langle w, a, c \rangle$ is a non-collider in $\mathcal{C}_k(D)$. Therefore, $\mathcal{C}_k(D')$ cannot be Markov equivalent to $\mathcal{C}_k(D)$, which is a contradiction. □

Now consider the execution of the algorithm $k$-PC . When the algorithm completes Step 4, from Corollary 4.2 we have that $K = $ PAG($\mathcal{C}_k(D)$). Since we start Step 5 with $K = PAG(\mathcal{C}_k(D))$, from Lemma A.19 and A.20, any arrowhead and tail orientation of the $K$ obtained at the end of step 5 must be consistent with the $k$-essential graph of $D$. Therefore, we have that $\varepsilon_k(D) \subseteq K$.

□

# B  FCI Orientation Rules

---

**Algorithm 2** FCI_Orient

---

**Input:** Mixed graph $K$
Apply the orientation rules of $\mathcal{R}1, \mathcal{R}2, \mathcal{R}3$ of [25] to $K$ until none applies.
Apply the orientation rules of $\mathcal{R}8, \mathcal{R}9, \mathcal{R}10$ of [25].
**Output:** $K$

---

We restate the FCI orientation rules in detail and demonstrate how they are applicable for learning $k$-closure graphs. The following definitions are from [25].

**Definition B.1** (Partial Mixed Graph (PMG)). Any graph that contains the edge marks arrowhead, tail, circle is called a partial mixed graph (PMG).

**Definition B.2** (Uncovered path). In a PMG, a path $\langle u_1, u_2 \ldots u_m \rangle$ is called an uncovered path if $u_i, u_{i+2}$ are non-adjacent for all $i \in \{1, 2, \ldots . m - 2\}$.

**Definition B.3** (Potentially directed path). In a PMG, a path $\langle u_1, u_2 \ldots u_m \rangle$ is called a potentially directed (p.d.) path if the edge between $u_i$ and $u_{i+1}$ does not have an arrowhead at $u_i$ for all $i \in \{1, 2, \ldots . m - 1\}$.

**Definition B.4** (Circle path). In a PMG, a path $\langle u_1, \ldots u_m \rangle$ is called a circle path if $u_i o\text{—}o u_{i+1}$ for all $i \in \{1, 2, \ldots . m - 1\}$.

Note that circle paths are special cases of p.d. paths.

Rules $1, 2, 3$ are straightforward extensions of the orientation rules for constraint-based learning to mixed graphs. For completeness, we restate them below. The star marks that appear both before and after the application of the rules are edge marks that remain unchanged by the rule.

$\mathcal{R}1$: If $a*\!\!\to bo\text{—}* c$, and $a, c$ are not adjacent, then orient the triple as $a*\!\!\to b \to c$.

$\mathcal{R}2$: If $a \to b*\!\!\to c$ or $a*\!\!\to b \to c$ and $a *\text{—}oc$, then orient $a *\text{—}oc$ as $a*\!\!\to c$.

$\mathcal{R}3$: If $a*\!\!\to b \leftarrow *c$, $a *\text{—}odo\text{—}* c$, $a, c$ are non-adjacent, and $d *\text{—}ob$ then orient $d *\text{—}ob$ are $d*\!\!\to b$.

We now restate FCI+ rules[3] $\mathcal{R}8, \mathcal{R}9, \mathcal{R}10$ and explain their relevance for learning $k$-closure graphs. Note that the rules are simplified since we do not have undirected edges that represent selection bias, and our undirected edges are treated as if they are circle edges.

$\mathcal{R}8$:   If $a \to b \to c$ and $ao\!\!\to c$, orient $ao\!\!\to c$ as $a \to c$.

$\mathcal{R}9$:   If $ao\!\!\to c$, and $p = \langle a, b, u_1, u_2 \ldots u_m, c \rangle$ is an uncovered p.d. path from $a$ to $c$ such that $b, c$ are non-adjacent, then orient $ao\!\!\to c$ as $a \to c$.

$\mathcal{R}10$:   Suppose $ao\!\!\to c, b \to c, d \to c$, $p_1$ is an uncovered p.d. path from $a$ to $d$ and $p_2$ is an uncovered p.d. path from $a$ to $b$. Let $t_d$ be the node adjacent to $a$ on $p_1$ ($t_d$ can be $d$) and $t_c$ be the node adjacent to $a$ on $p_2$ ($t_c$ can be $c$). If $t_d, t_c$ are distinct and non-adjacent, then orient $ao\!\!\to c$ as $a \to c$.

$\mathcal{R}11$ and $\mathcal{R}12$ cannot replace any of the above rules. For example, consider Figure 5. None of the FCI rules apply to the output of Step 3, thus we can only learn of the unshielded colliders at the end of Step 4 of the algorithm. The completeness of FCI implies that any edge $xo\!\!\to y$ can be oriented as $x \leftrightarrow y$ or $x \to y$ and give a MAG consistent with the PAG. However, not all such MAGs are valid $k$-closure graphs. $\mathcal{R}11$ can be applied to orient several tails, which gives the graph in $(d)$. Similarly in Figure 6, $\mathcal{R}12$ helps orient the tail edges between $a, b$ which cannot be learned by FCI rules.

---

[3]This version was originally called A-FCI, short for augmented FCI rules by [25]. Augmented graphs are recently used in a different context in the causality literature, which is why in this work we are calling this version FCI+ to avoid confusion.

# C  Sample Runs of $k$-PC Algorithm

Consider the figures below for two sample runs of $k$-PC algorithm. Note that $k$-PC outputs the $k$-essential graph in these examples, i.e., it can orient every invariant arrowhead and tail mark in the $k$-closure graph of $D$.

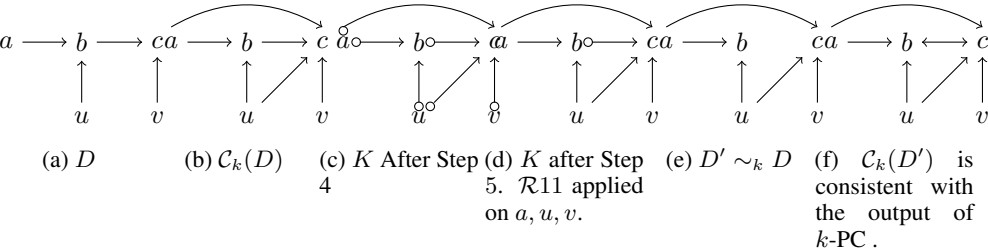

(a) $D$          (b) $\mathcal{C}_k(D)$          (c) $K$ After Step 4          (d) $K$ after Step 5. $\mathcal{R}11$ applied on $a, u, v$.          (e) $D' \sim_k D$          (f) $\mathcal{C}_k(D')$ is consistent with the output of $k$-PC .

Figure 5: An example where $\mathcal{R}11$ helps orient tails. ($a$) A DAG $D$. ($b$) $k$-closure graph of $D$ for $k = 0$. ($c$) $K$ after Step 4 of k$-PC$, the same as $PAG(\mathcal{C}_k(D))$. ($d$) $\mathcal{R}11$ helps orient several tail edges. ($e$) A DAG $D'$ that is $k$-Markov to $D$. ($f$) $k$-closure graph of $D'$, which is Markov equivalent to $\mathcal{C}_k(D)$, showing that the circle at $bo\to c$ is not an invariant tail. Thus $k$-PC outputs $k$-essential graph $\varepsilon_k(D)$ in this case.

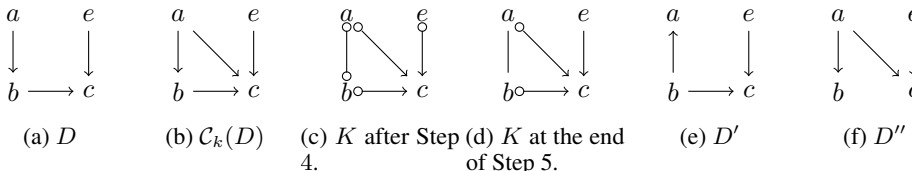

(a) $D$          (b) $\mathcal{C}_k(D)$          (c) $K$ after Step 4.          (d) $K$ at the end of Step 5.          (e) $D'$          (f) $D''$

Figure 6: ($a$) A causal graph $D$. ($b$) $k$-closure of $D$ for $k = 0$. ($c$) $K$ at the end of Step 4. ($d$) Node $a$ has one edge $ao$—$o$, $ao$—$ob$. Thus we have $\mathcal{C} = \mathcal{C}^*$ since $b$ is non-adjacent to any other nodes in $C$ since there are no other nodes in $C$. Thus it is oriented as $a$—$b$ due to $\mathcal{R}12$. $e \to c$ is oriented due to $\mathcal{R}11$, similarly since $\mathcal{C} = \emptyset$ and the node $c$ is trivially non-adjacent to all nodes in $C$. ($e, f$) $D', D''$ are $k$-Markov equivalent to $D$ and their $k$-closure graphs contain $a \leftrightarrow c$ and $b \leftrightarrow c$, respectively. This shows that the graph in ($d$) given by $k$-PC is the $k$-essential graph $\varepsilon_k(D)$.

# D   Discussions

## D.1   No Local $k$-Markov Equivalence Characterization on DAG Space

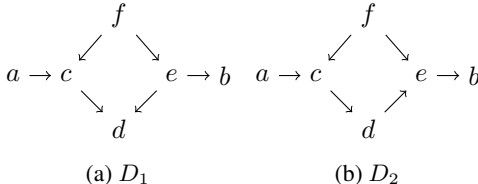

(a) $D_1$        (b) $D_2$

Figure 7: As a collider, $d$ blocks the path $(a, c, d, e, b)$ in $D_3$, but not in $D_4$. Accordingly, $(a \perp\!\!\!\perp b)_{D_1}$ $(a \not\perp\!\!\!\perp b)_{D_2}$ and $D_1, D_2$ are not $k$-Markov for $k = 0$. However, this does not appear as a local condition since $c, e$ are not separable by conditioning sets of size up to 0 in both graphs. Therefore, a local characterization of equivalence like Verma and Pearl is not possible when we are only allowed to check degree-$k$ d-separation tests.

Consider the graphs in Figure 7. The only difference is the orientation of the edge between $e, d$. Due to the collider $c \rightarrow d \leftarrow e$, we have $a \perp\!\!\!\perp b$ in $D_1$ but not in $D_2$. Therefore for $k = 0$, we have that $D_1, D_2$ are not $k$-Markov equivalent. However, this is not detectable locally: The endpoints of the collider responsible for the change of the $k$-Markov equivalence class cannot be d-separated. The effect of the collider can be detected only farther out in the graph between $a, b$. This shows that a local characterization similar to Theorem 2.10 is not possible for $k$-Markov equivalence.

## D.2   $k$-closure Graphs vs. MAGs

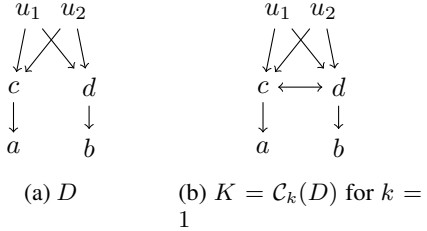

(a) $D$        (b) $K = \mathcal{C}_k(D)$ for $k = 1$

Figure 8: The mixed graph on the right is a valid $k$-closure graph for $k = 1$: It is the $k$-closure graph of the DAG in $(a)$. However, it is not a valid $k$-closure graph for $k = 2$. Because after removing $c \leftrightarrow d$, it is possible to d-separate $c, d$ by conditioning on $u_1, u_2$.

In this section, we give an example for a MAG that is not a valid $k$-closure graph. Consider the graph in Figure 8. $K$ in $(b)$ is a valid $k$-closure graph for $k = 1$ since it is the $k$-closure graph of $D$. However, $K$ is not a valid $k$-closure graph for $k = 2$. This is because the bidirected edge $c \leftrightarrow d$ is added between a pair that is not $k$-covered for $k = 2$: We have $c \perp\!\!\!\perp d \,| \, u_1, u_2$ in $D$. In fact, using the if and only if characterization in Theorem 3.9, we can show that there does not exist any $D'$ with the given $k$-closure graph where $c, d$ have a bidirected edge.

## D.3   Bidirected Edge in $k$-essential Graphs

In Figure 9, since every endpoint is an arrowhead and is part of an unshielded collider, there is no other Markov equivalent $k$-closure graphs, which implies that the $k$-essential graph is the same as the $k$-closure graph. Thus, the edge $c \leftrightarrow e$ is in $\varepsilon_k(D)$. This example shows that we can learn that two nodes do not cause each other using *conditional independence tests that are not even powerful enough to make them conditionally independent*. It is worth noting that LOCI [23] can also infer this fact by removing this edge.

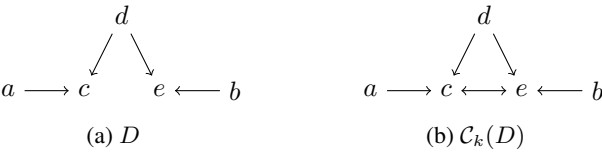

(a) $D$                         (b) $\mathcal{C}_k(D)$

Figure 9: A DAG with a size-1 $k$-Markov equivalence class for $k = 1$. Observe that $\mathcal{C}_k(D)$ only has unshielded colliders and thus there is no other $k$-closure graph that is Markov equivalent. Thus, this $k$-closure is at the same time the $k$-essential graph of $D$ and can be learned from data. In this case, we can learn $c, e$ do not cause each other despite not being separable in the data.

## D.4   $k$-PC is Incomplete

One might hope that $k$-PC is complete and outputs the $k$-essential graph $\varepsilon_k(D)$. This, however, is not true. We discuss an example where $k$-PC cannot orient an invariant tail mark.

First, observe that $k$-PC does not leverage the value of $k$. If we had an efficient way to answer the question "*Is there a $k$-closure graph that is consistent with $K$ in which $a, b$ are $k$-covered?"* then we could leverage this to orient more $o\!\rightarrow$ edges as $\rightarrow$ edges. As an example, consider the causal graph in Figure 10. $d$ may have an incoming edge that prevents us from eliminating the possibility that $do\!\rightarrow b$ is a bidirected edge $d \leftrightarrow b$. However, we can only have a single d-connecting path between $d, b$. Since $d$ is a non-collider along $a$—$d$—$c$, one of the edges must be out of $d$. Suppose $d \leftarrow a$ and $d \rightarrow c$. For $c$ to not block this path it has to be a non-collider, and thus $c \rightarrow b$. This is the only way we can have two d-connecting paths between $d, b$ in the underlying DAG. However, now we have the path $d \rightarrow c \rightarrow b$, which makes $d$ an ancestor of $b$. Therefore the edge $d \leftrightarrow b$ is inconsistent. Thus in $k$-essential graph of $D$, we must have a tail at $d$ as $d \rightarrow b$. This cannot be learned by $k$-PC .

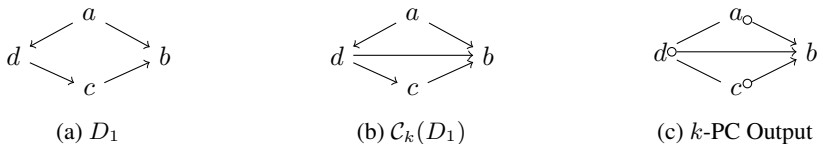

(a) $D_1$                         (b) $\mathcal{C}_k(D_1)$                         (c) $k$-PC Output

Figure 10: A graph where $k$-essential graph contains an invariant tail that cannot be learned by $k$-PC algorithm. $k = 1$; thus $d, b$ are $k$-covered but not $a, c$, which gives the $k$-closure graph on the right. $k$-PC algorithm outputs the graph in $(c)$. However, there is no $k$-closure in the Markov equivalence class where $d \leftrightarrow b$. Thus the edge in the $k$-essential graph should be $d \rightarrow b$. Similarly, $ao\!\rightarrow b$ should be $a \rightarrow b$, however this requires reasoning about the number of d-connecting paths between $a, b$ in $K$ that is not captured in $k$-PC .

We observe that a local algorithm such as $k$-PC cannot be used to assess if there is some $k$-closure graph that is consistent with the current graph in which two nodes are $k$-covered without the edge between them. One practical strategy would be to take the output of the $k$-PC algorithm and list all MAGs consistent with the circle edges, and then check if they are valid $k$-closure graphs by pruning every edge using Theorem 3.9. For small or sparse $k$-closure graphs, $k$-PC could be a practical way to reduce the search space efficiently, and then we can conduct an exhaustive search as the next step to obtain a sound and complete algorithm for learning the $k$-essential graph.

## D.5   Heuristic Uses of Bounded Size Conditioning Sets

For large number of variables, and except for very sparse graphs, constraint-based algorithms take significant time to complete. This is because the progressive nature of such tests is not able to sufficiently sparsify the graph with low-degree conditional independence tests, and they have to perform many tests: If the neighbor size is $\mathcal{O}(n)$, where $n$ is the number of nodes, algorithm needs to check exponentially many subsets of nodes in the conditioning set.

To prevent this issue, several implementations of these algorithms have the added functionality to restrict this search by limiting the size of the conditioning set. For example, causal-learn package has this functionality. However, this is a heuristic that simply prematurely stops the search algorithms.

The results of our paper build the theoretical understanding of what is learnable in this setting with a new equivalence class and its graphical representation.

## D.6 Comparison of $k$-PC Output to LOCI Output

**Lemma D.1.** *Consider three nodes $a, b, c$. Suppose $a \not\perp\!\!\!\perp b \,|\, S$ , $b \not\perp\!\!\!\perp c \,|\, S$ , $a \perp\!\!\!\perp c \,|\, S$ for some set $S : |S| \leq k$ and $b \notin S$. If $b, c$ are $k$-covered , then $k$-PC orients the edge between $b, c$ as $b \leftarrow\!\!*c$.*

*Proof.* In the following, we show that given the CI pattern that LOCI uses to orient edges, $k$-PC also orients the same edges. Consider the CI pattern that LOCI uses between three nodes $a, b, c$.

Conditioned on $S$, there is a d-connecting path between $a, b$; let us call this path $p$. Conditioned on $S$, there is a d-connecting path between $b, c$; let us call this path $q$. Consider the path obtained by concatenating $p, q$. Since this path must be d-separating, $b$ must be a collider on it and it must be the case that $b \notin An(S)$. Let the node that is adjacent to $b$ along $p$ be $a'$ and the node adjacent to $b$ along $q$ be $c'$. Thus we have $a' \rightarrow b \leftarrow c'$.

***Case 1:*** Now suppose that $a'$ and $c$ are separable by some $T : |T| \leq k$. $k$-PC would then orient $a'*\!\!\rightarrow b \leftarrow\!\!*c$ since $b, c$ remains adjacent throughout the execution of $k$-PC . Thus $b \leftarrow\!\!*c$ would be oriented in this case.

***Case 2:*** Suppose that $a', c$ are $k$-covered . Then, given $S$, there is a d-connecting path between $a', c$. Now consider the path obtained by concatenating the subpath of $p$ between $a, a'$ and this d-connecting path. Since $a, c$ are d-separated given $S$, $a'$ must be a collider along this path and a non-ancestor of $S$. Thus we must have $a'' \rightarrow a' \rightarrow b$ as the last three nodes of path $p$.

***Case 2.a:*** Suppose $a'', c$ are separable by some $T : |T| \leq k$. Since $a', c$ are $k$-covered , they remain adjacent throughout the execution of $k$-PC . Thus, $k$-PC would orient $a''*\!\!\rightarrow a' \leftarrow\!\!*c$. Now we consider two sub-cases:

***Case 2.a.i:*** Suppose $a'', b$ are separable by some set of size at most $k$. Then, $a''*\!\!\rightarrow a' - b$ is an unshielded triple and it would be oriented by the first Meek rule of $k$-PC as $a''*\!\!\rightarrow a' \rightarrow b$. Now we have the following edges oriented: $b \leftarrow\!\!*a' \leftarrow\!\!*c$, and $b, c$ adjacent. By the second Meek rule, $k$-PC will then orient $b \leftarrow\!\!*c$. This establishes that in this sub-case, $k$-PC would also orient the arrowhead adjacent to $b$ for the edge between $b, c$.

***Case 2.a.ii:*** Now consider the second sub-case: Suppose $a'', b$ are $k$-covered . Thus, throughout the execution of $k$-PC , $a'', b$ are adjacent. In this case, $a''o$—$obo$—$oc$ forms an unshielded collider and the algorithm would orient them as $a''*\!\!\rightarrow b \leftarrow\!\!*c$ due to the independence statement $a'' \perp\!\!\!\perp c \,|\, T$. Therefore, the arrowhead at $b$ would be oriented in this case as well.

***Case 2.b:*** Now suppose $a'', c$ are $k$-covered . Thus, there must be a d-connecting path between $a'', c$ given $S$. Now consider the path obtained by concatenating the subpath of $p$ between $a, a''$ and this d-connecting path. Since $a \perp\!\!\!\perp c \,|\, S$, it must be that $a''$ is a collider along this path and that $a'' \notin An(S)$. Thus, along this path we have $a''' \rightarrow a'' \leftarrow \ldots$. Therefore, the last four nodes of the path $p$ is $a''' \rightarrow a'' \rightarrow a' \rightarrow b$, and $a', c$ are $k$-covered , $a'', c$ are $k$-covered .

***Case 2.b.i:*** Now suppose $a''', c$ are separable by some $T : |T| \leq k$.

***Case 2.b.i.$\alpha$:*** If the pair $a''', b$ is $k$-covered , following the above argument, we would orient $a'''*\!\!\rightarrow b \leftarrow\!\!*c$ with the statement $a''' \perp\!\!\!\perp c \,|\, S$, and the arrowhead adjacent to $b$ along the edge between $b, c$ would have been oriented.

***Case 2.b.i.$\beta$:*** Suppose $a''', b$ pair is separable.

***Case 2.b.i.$\beta$.1:*** Suppose $a''', a'$ is $k$-covered . We would orient $a'''*\!\!\rightarrow a' \leftarrow\!\!*c$ due to the statement $a''' \perp\!\!\!\perp c \,|\, S$. And since $a''', b$ are not $k$-covered , Meek rule one would orient the subgraph $a'''*\!\!\rightarrow a'o$—$ob$ as $a'''*\!\!\rightarrow a'*\!\!\rightarrow b$. Since $b \leftarrow\!\!*a' \leftarrow\!\!*c$ and $bo$—$oc$, second Meek rule would orient the arrowhead at $b$ along the edge between $b, c$.

***Case 2.b.i.$\beta$.2:*** Finally, suppose $a''', b$ and $a''', a'$ are both separable. Now $k$-PC applies Meek rule one directly to orient $a'''*\!\!\rightarrow a''*\!\!\rightarrow a'$. Now that we have $a' \leftarrow\!\!*a'' \leftarrow\!\!*c$, and that $a', c$ are adjacent, Meek rule two will orient $a' \leftarrow\!\!*c$. Another application of Meek rule one would give $a''*\!\!\rightarrow a'*\!\!\rightarrow b$ and now since we have $b \leftarrow\!\!*a' \leftarrow\!\!*c$ and that $b, c$ are adjacent, Meek rule two would orient $b \leftarrow\!\!*c$.

***Case 2.b.j and beyond:*** Finally, if $a'''$, $c$ are $k$-covered , following a similar argument, either the node adjacent to $a'''$ along $p$ (towards $a$) is separable with $c$, in which case following a similar argument as above would orient $b \leftarrow\!* c$, or we continue until $a, c$ become adjacent. The latter cannot happen since that contradicts with the fact that $a \perp\!\!\!\perp c \,|\, S$. Thus, there must exist some node $u$ along $p$ that is separable from $c$, and the subpath of $p$ between $u, b$ is directed. Following the argument above, repeated application of Meek rules one and two will result in the orientation of the edge between $b, c$ as $b \leftarrow\!* c$. This establishes that $k$-PC orients at least as much arrowheads as LOCI. $\qquad\square$

The corollary of this lemma is that $k$-PC orients all arrowheads oriented by LOCI:

**Corollary D.2.** *Any arrowhead oriented by LOCI is also oriented by $k$-PC .*

*Proof.* Suppose $a \not\perp\!\!\!\perp b \,|\, S$ , $b \not\perp\!\!\!\perp c \,|\, S$ , $a \perp\!\!\!\perp c \,|\, S$ for some set $S : |S| \leq k$ and $b \notin S$. Observe that if $a, b$ and $b, c$ are $k$-covered then $k$-PC would orient the edges between them as $a*\!\!\rightarrow b \leftarrow\!* c$ due to the independence statement $a \perp\!\!\!\perp c \,|\, S$. Moreover, if $a, b$ and $b, c$ are both separable by some sets of size at most $k$, then both LOCI and $k$-PC would make $a, b$ and $b, c$ non-adjacent and thus neither algorithm orients an edge between them. Therefore, the only non-trivial case is when only one of the two pairs is $k$-covered . For this case, since the pre-condition of Lemma D.1 is identical to the condition of LOCI to orient any edge.

LOCI applies the three Meek rules after orienting these arrowhead marks. Since $k$-PC repeatedly applies a set of Meek rules that include these three rules, $k$-PC orients at least as many arrowheads as LOCI. Thus, the corollary follows. $\qquad\square$

Next, we show that they both carry the same adjacency information.

**Corollary D.3.** *Any pair that is non-adjacent in LOCI output is either non-adjacent, or adjacent via a bidirected edge in $k$-PC output.*

*Proof.* LOCI makes a pair non-adjacent in two ways. If LOCI makes a pair non-adjacent since they are separable, $k$-PC also will make them non-adjacent. Suppose LOCI makes a pair $a, b$ non-adjacent due to the following CI pattern, which are otherwise $k$-covered : $u \not\perp\!\!\!\perp a \,|\, S_1$ , $a \not\perp\!\!\!\perp b \,|\, S_1$ , $u \perp\!\!\!\perp b \,|\, S_1$ and $a \not\perp\!\!\!\perp b \,|\, S_2$ , $b \not\perp\!\!\!\perp v \,|\, S_2$ , $a \perp\!\!\!\perp v \,|\, S_2$ for some $u, v, S_1, S_2$. Note that by Lemma D.1, $k$-PC marks both endpoints of the edge between $a, b$ as arrowheads. This concludes the proof. $\qquad\square$

These results can be combined with the example in Figure 2 to show that the $k$-PC output carries strictly more information about the set of causal graphs that entail the set of degree-$k$ d-separation statements than the output of LOCI, even though $k$-PC is not complete for learning our equivalence class as discussed in Section D.4.

### D.7 Comparison of $k$-PC Output to AnytimeFCI Output

The undirected edges in our representation do not represent selection bias. We use them to represent a different graph union operation. The lack of latents gives us this flexibility, which allows us to distinguish different sets of graphs by using both — and $o$—$o$ edge marks for this.

For a concrete example on why AnytimeFCI is less informative than $k$-PC in our setting, consider the causal graph in Figure 2. Here, $k$-PC will learn the representation on the right: We can infer that either $a$ causes $d$ or $d$ causes $a$, noted by the undirected edge $a$ — $d$ in our representation, rather than the circle edge $o$—$o$, which would allow $a, d$ to be non-adjacent in some DAGs. Anytime FCI here will instead output circle edges between $a$ and $d$, i.e., $ao$—$od$. Even if we try to incorporate the knowledge that the underlying graph is causally sufficient, it is not obvious how one can conclude that $a$ and $d$ must be adjacent in *every DAG that induces the same degree-$k$ d-separation statements* from the Anytime FCI output. One way to do this would be to remove the edge from this PAG and check whether $a, d$ can now be d-separated by some conditioning set of size at most $k$. If yes, this changes the equivalence class, which means the edge must be there in any DAG. Our local rule $\mathcal{R}12$ can instead orient this as an undirected edge, signifying that $a$ and $d$ must be adjacent in every underlying DAG by leveraging our equivalence class representation, without having to run this type of post-processing which might be computationally intensive for larger graphs.

# E    Additional Experiments

## E.1    Experiments vs. LOCI

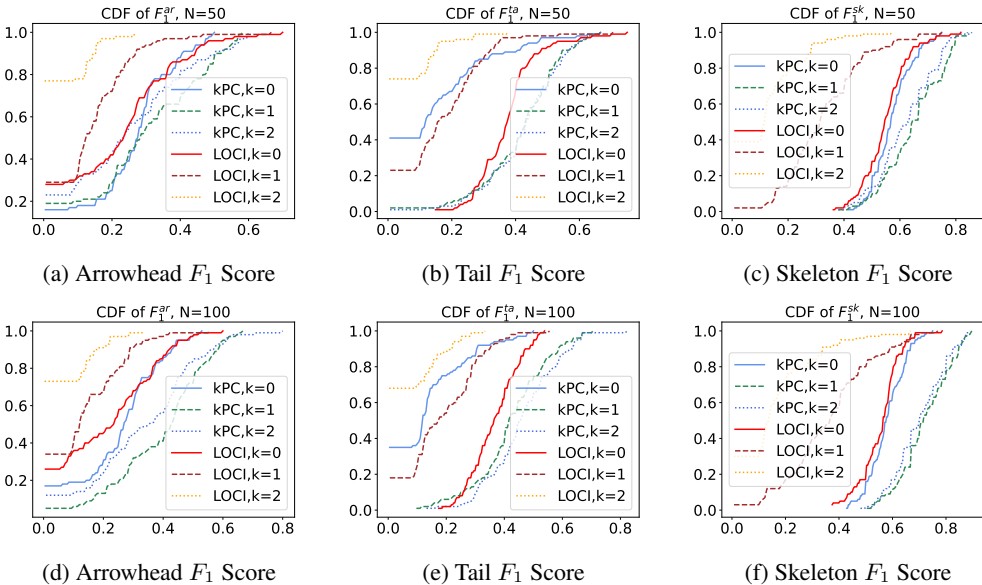

Figure 11: Empirical cumulative distribution function of various $F_1$ scores on 100 random DAGs on 10 nodes. We pick the parents of each node in a fixed total order randomly so that the expected value of parents is 3. The conditional distribution of every node given any configuration of its parent set is sampled independently, uniformly randomly from the corresponding dimensional probability simplex. Three datasets are sampled per instance. Performance of $k$-PC and LOCI [23] are similar as expected. We still observe a similar trend as PC that arrowhead score of $k$-PC is better. For tail accuracy, the result depends on the value of $k$. For small $k$, LOCI outperforms $k$-PC whereas for larger $k$, $k$-PC outperforms LOCI in the small sample regime. Different from Figure 3, we compare the outputs to the true DAG instead of essential graph since no algorithm in this comparison can achieve essential graph.

## E.2 Experiments vs. NOTEARS

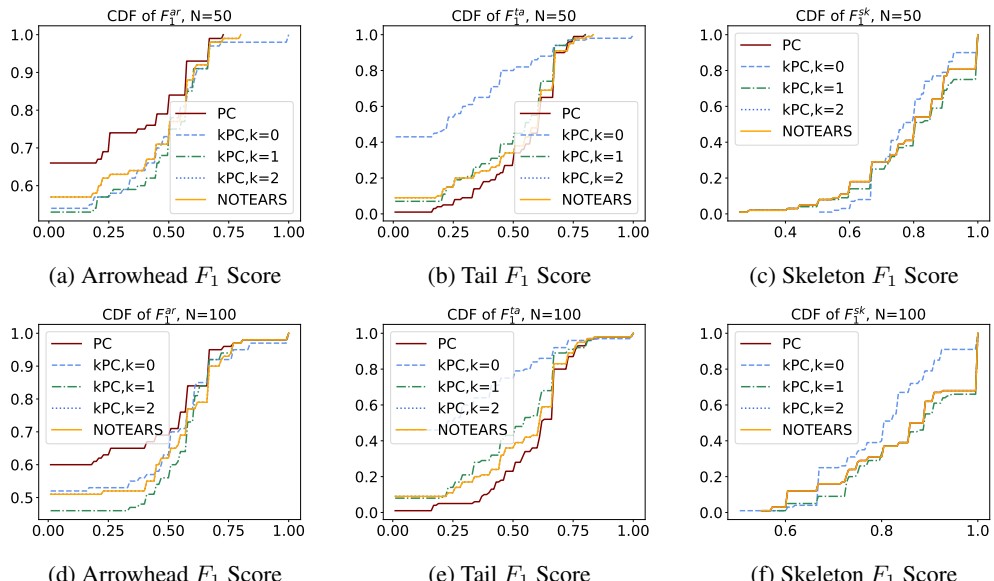

(a) Arrowhead $F_1$ Score     (b) Tail $F_1$ Score     (c) Skeleton $F_1$ Score

(d) Arrowhead $F_1$ Score     (e) Tail $F_1$ Score     (f) Skeleton $F_1$ Score

Figure 12: Empirical cumulative distribution function of various $F_1$ scores on 100 random DAGs on 5 nodes. For each DAG, a linear SCM is sampled as follows: Each coefficient is chosen randomly in the range $[-3, 3]$. Exogenous noise terms are jointly independent unit Gaussian. Performance of $k$-PC vs. NOTEARS [28]. We observe a similar trend as PC. NOTEARS is slightly better than PC consistently. Despite this, $k$-PC outperforms both in the low-sample regime. Metrics are computed against the true DAG.

## E.3  More Experiments vs. PC

In this section, we show a larger range of $N$ (number of samples). We also explore the behaivor for graphs with different edge densities and higher number of nodes (10).

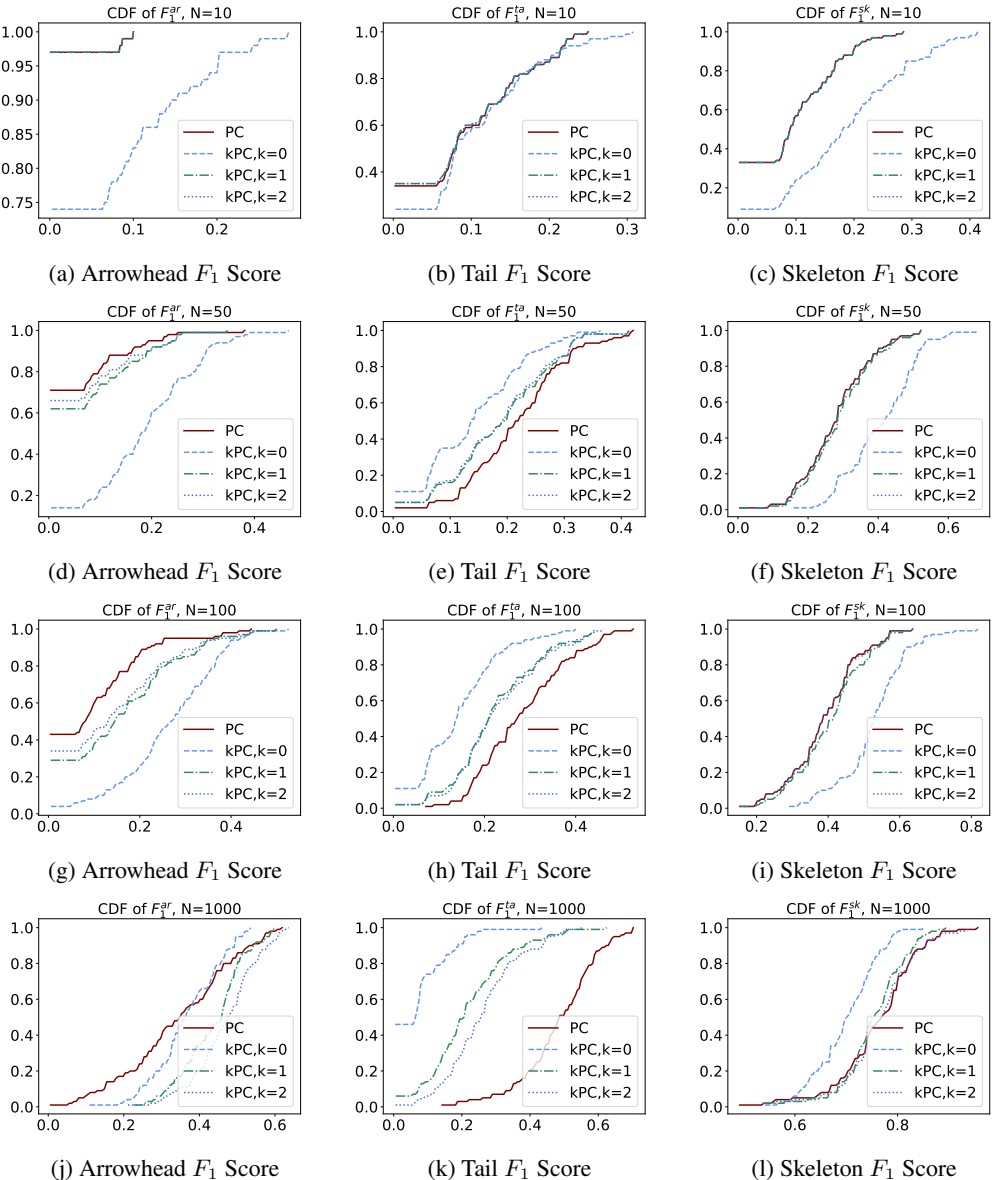

(a) Arrowhead $F_1$ Score

(b) Tail $F_1$ Score

(c) Skeleton $F_1$ Score

(d) Arrowhead $F_1$ Score

(e) Tail $F_1$ Score

(f) Skeleton $F_1$ Score

(g) Arrowhead $F_1$ Score

(h) Tail $F_1$ Score

(i) Skeleton $F_1$ Score

(j) Arrowhead $F_1$ Score

(k) Tail $F_1$ Score

(l) Skeleton $F_1$ Score

Figure 13: Empirical cumulative distribution function of various $F_1$ scores on 100 random DAGs on 10 nodes. For each DAG, conditional probability tables are independently and uniformly randomly filled from the corresponding probability simplex. Three datasets are sampled per instance. The lower the curve the better. The maximum number of edges is 30. Even in the extreme case of just 10 samples (10 node-graphs), $k$-PC for $k = 0$ provides improvement to all scores. $k$ should be gradually increased as more samples are available to make best use of the available data. For example, for 1000 samples, $k = 2$ provides the best arrowhead score while not giving up as much tail score as $k = 0$.

Next, we present combined metrics for this same setup. Namely, we show the advantage of our algorithm in terms of the sum of arrowhead and tail $F_1$ scores, and the sum of arrowhead, tail and skeleton $F_1$ scores.

## E.4 Experiments of Section E.3

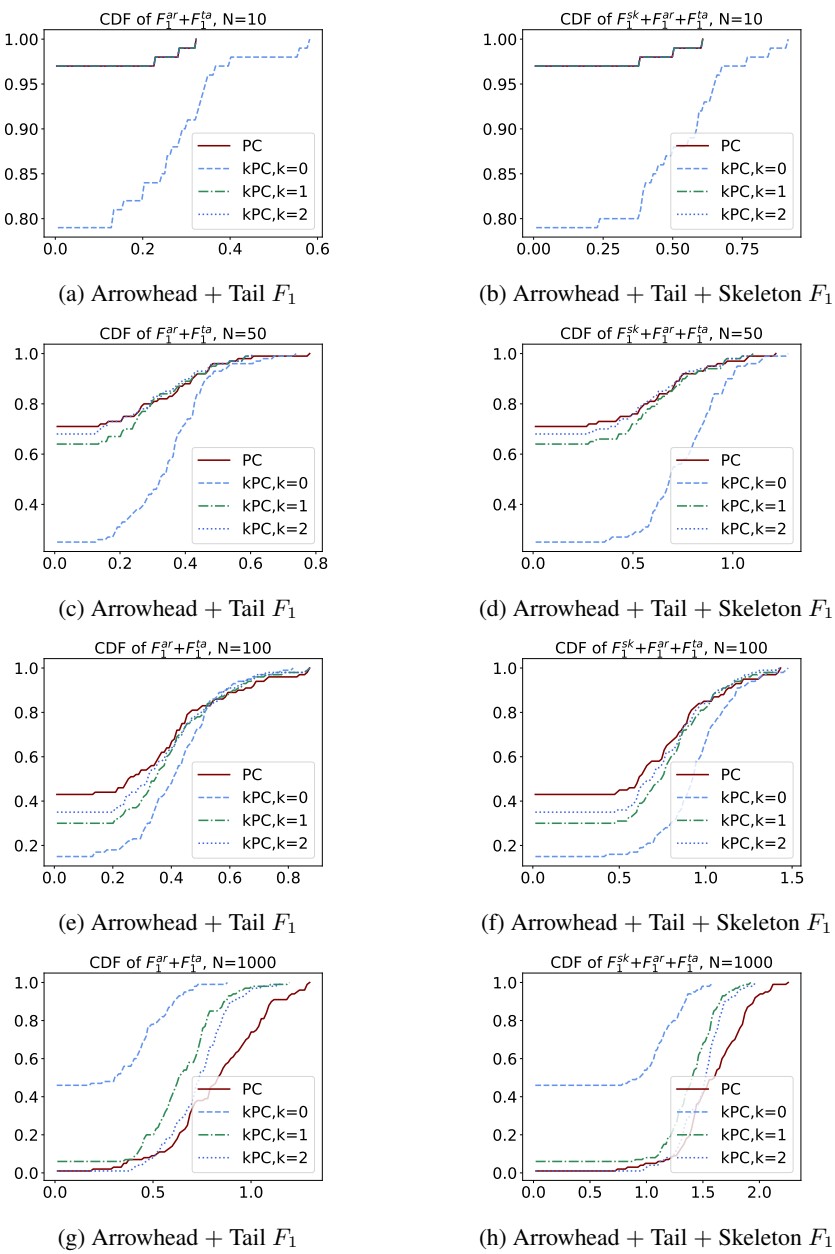

Figure 14: Results of the experiments in Section E.3 in terms of combined scores. For each instance, arrowhead and tail $F_1$ scores are added before computing CDFs on the left. On the right, arrowhead, tail and skeleton $F_1$ scores are added together.

## E.5 Experiments vs. Conservative PC

We use pcalg package in R as the implementation for conservative PC [14].

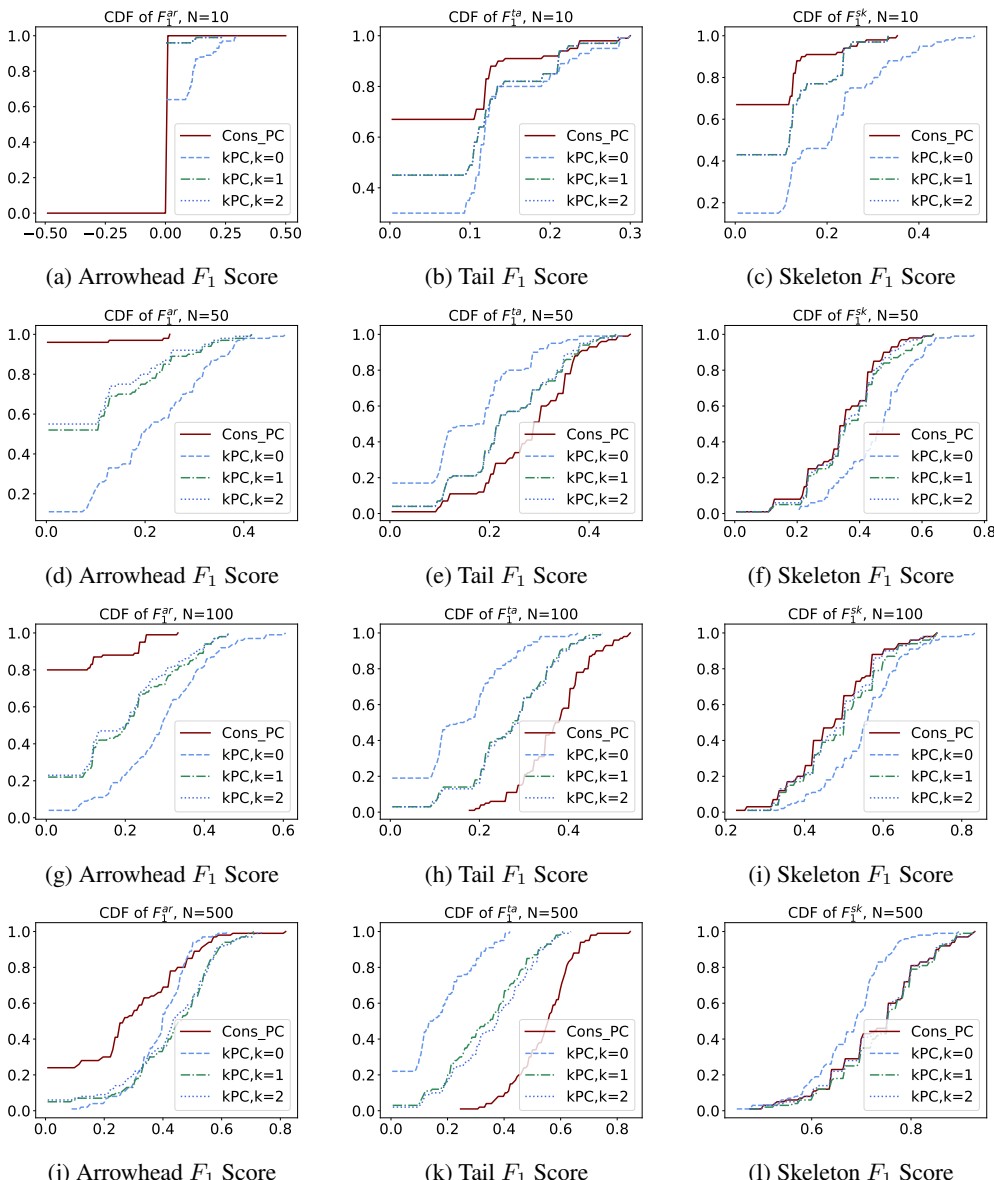

(a) Arrowhead $F_1$ Score

(b) Tail $F_1$ Score

(c) Skeleton $F_1$ Score

(d) Arrowhead $F_1$ Score

(e) Tail $F_1$ Score

(f) Skeleton $F_1$ Score

(g) Arrowhead $F_1$ Score

(h) Tail $F_1$ Score

(i) Skeleton $F_1$ Score

(j) Arrowhead $F_1$ Score

(k) Tail $F_1$ Score

(l) Skeleton $F_1$ Score

Figure 15: Empirical cumulative distribution function of various $F_1$ scores on 100 random DAGs on 10 nodes. For each DAG, conditional probability tables are independently and uniformly randomly filled from the corresponding probability simplex. One dataset is sampled per instance. The lower the curve the better. The maximum number of edges is 15. $k$-PC maintains an advantage against conservative PC in the arrowhead and skeleton $F_1$ scores in the low-sample regime.

