# OpenReview forum: "Characterization and Learning of Causal Graphs with Small Conditioning Sets"
_NeurIPS.cc/2023/Conference — NeurIPS 2023 poster_

### Official Review · Reviewer_mhM7 · 2023-06-27

**Soundness:** 4 excellent
**Presentation:** 3 good
**Contribution:** 3 good
**Rating:** 7
**Confidence:** 4

**Summary:**

Besides computational complexity, the weak spot of constraint-based causal discovery is the assumption of having an independence oracle. Especially with large conditioning sets, independence testing is difficult. This paper aims to address this limitation by restricting the number of conditioning variables and provides a sound framework that allows us to study the guarantees one can obtain for constraint-based structure learning if one restricts the size of the conditioning sets. The authors show that the results can contain bi-directed edges and relate the obtained graph to maximal ancestral graphs. They further provide a sound algorithm to learn such graphs and evaluate it on a few synthetic and semi-synthetic examples.

The theoretical contributions of the paper are interesting and presented clearly to most parts. The main limitation of the work are the points that I mention with respect to the empirical evaluation.


**Strengths:**

- Rigorous definition of assumptions, goal, and solution that can be obtained under the concept of restricting the size of the conditioning set.
- The new formulation of k-closed graphs was introduced well and discussed in the context of previous notions (ancestral graphs). Further, Markov equivalence and faithfulness are adapted for the introduced graphs.
- The authors propose a learning algorithm k-PC to learn the Markov equivalence class of the introduced k-closure of the true DAG.
- The proposed algorithm has been empirically evaluated to PC, LOCI and NOTEARS, and the code has been provided for reproducibility.


**Weaknesses:**

- The performance gain seems to be only evident in small sample regimes.
- The new approach does not clearly outperform LOCI & it would be interesting to see how PC performs when restricting the conditioning set, as well.
- It would be interesting to study a real-world example, or DAGs with non-linear relations. I could imagine that on the latter, using a smaller conditioning set can be a more substantial advantage, as independence testing with large conditioning sets is especially difficult for arbitrary functional relationships. For more details on the problem and maybe a good source to motivate the proposed algorithm please consider Shah and Peters (2020)

Shah, Peters, The Hardness of Conditional Independence Testing and the Generalised Covariance Measure, The Annals of Statistics, 2020

Minor (suggestions):
- For easier comparison, it might be beneficial to compute the area under the curve for the metrics shown in Figure 3.

**Questions:**

- What is the intuition of step 5 in the algorithm? Why do we have to exclude it for Corollary 4.2? Not much intuition is provided regarding this step, and it just occurs in Corollary 4.2.
- I understand that if we just run the PC algorithm and restrict the conditioning set it does not inherit the same guarantees as PC-k, e.g., it would not find any bi-directed edges. What would be the other differences/failure cases of this naive approach?
- Assume one naively restricts the maximum size of the conditioning set in the PC algorithm, which I think is an option in the pcalg package, for example. How does this version of PC perform if we restrict the conditioning size in a similar manner as for the proposed method (e.g., for the experiments shown in Fig. 3 & 4)?
- How does the computational complexity or empirical runtime compare to PC?


**Limitations:**

Standard limitations occurring in constraint-based causal discovery (a variant of faithfulness, independence oracle).

---

> ### Author Rebuttal · Authors · 2023-08-06
>
> We would like to thank the reviewer for their detailed comments and suggestions. Please see below for our replies.
>
> **1.** "performance gain seems to be only evident in small sample regimes"\
> Note that showing our method outperforms the existing approaches in the small sample regime was the main goal of our experiments. When the sample size is large, k-PC will only be less informative than PC since high-degree CI tests become more reliable, and they carry information about the underlying causal structure.
>
> **2.** "what if we run PC, and early terminate? Any failure cases?"\
> This is a great question. Note that running PC only for conditioning sets of size at most k will lead to wrong causal conclusions. To see this, consider the graph in Figure 9 (in appendix), reproduced below for convenience:
>
> X->Y<-Z->U<-V
>
> Let k=0. Modifying PC to only condition on the empty set leaves in the spurious edge Y-U in the skeleton phase. Applying orientation rules afterwards, we can get Y<-U or Y->U depending on which unshielded collider orientation phase is conducted, either X->Y<-U or Y->Z<-T. Both of these would be wrong. Thus, basic PC is not sound if we simply do not condition on certain subsets. Alternative versions such as stable PC might detect these incompatibilities, but they don't have a systematic way to handle them except not orienting such edges. Our algorithm orients as bidirected, which allows inferring them as spurious edges.
>
> Since PC with this kind of early stopping is not sound even in the infinite sample regime, we have not included it in our experiments.
>
> **3.** "DAGs with nonlinear relations?"\
> Apologies for not being clear on this: Please note that most of our experiments are with non-parametric relations, i.e., with discrete variables with randomly chosen conditional probability tables (uniformly from the probability simplex). All discrete variables are binary since the results did not change with larger support and we were able to scale graphs easier with binary nodes. Note that we use linear models only while comparing our method against NOTEARS, whose baseline version uses the linearity assumption. We will clarify this in the manuscript.
>
> **4.** "Shah and Peters (2020)"\
> Thank you for this pointer! We will add this citation to help motivate our work from the hardness of independence testing.
>
> **5.** "Corollary 4.2 and Step 5 of Alg."\
> Corollary 4.2 says that we can learn as much as the PAG of our representation. And this is sound and complete. However, to learn more about the representation (k-essential graph), we need to have more orientation rules. These arise since our k-essential graph is not any ancestral graph, but has more structure. In other words, there is more to be learned beyond PAG. The rules in Step 5 allow us to learn more about the k-essential graph beyond the FCI orientation rules. We show that the overall algorithm with these rules is sound in Theorem 4.4. For a discussion on the challenges of completeness for learning the k-essential graph, please see the discussion in Section E.2 (appendix).
>
> **6.** "Computational complexity vs PC"\
> The complexity of the learning algorithm will be similar to an early-stopped version of FCI, called Anytime FCI by Spirtes 2001. Although this is complicated and would depend on other parameters in the graph, such as primitive inducing paths, and thus the number and location of unobserved confounders, we can also roughly bound this by $\mathcal{O}(n^{k+2})$ since for any pair, we will not be searching for separating sets beyond the $\mathcal{O}(n^k)$ subsets of size at most $k$. Further runtime improvements, such as RFCI by Colombo et al. 2012 might be possible, but this requires a further understanding of the structure of k-closure graphs, which might be an interesting future direction. PC on the other hand runs roughly in time $\mathcal{O}(n^d)$ where $d$ is the degree of the graph. Thus, for sparse graphs and small $k$, we can expect a similar runtime.
>
> We will add this discussion with the relevant citations below to the manuscript.
>
> [Spirtes] "An Anytime Algorithm for Causal Inference," 2001.
>
> [Colombo, Maathuis, Kalisch and Richardson] "Learning High-Dimensional Directed Acyclic Graphs With Latent And Selection Variables," 2012.
>
> Thank you very much for your feedback and valuable comments! We hope that our responses will give the reviewer more confidence in their acceptance recommendation and that they would consider increasing their score.

---

> > ### Comment · Reviewer_mhM7 · 2023-08-15
> >
> > I would like to thank the authors for their clarifications, which could alleviate some of my confusion regarding the experiments. I will increase my score to “accept”.

---

### Official Review · Reviewer_Gg3j · 2023-07-05

**Soundness:** 4 excellent
**Presentation:** 4 excellent
**Contribution:** 3 good
**Rating:** 7
**Confidence:** 3

**Summary:**

A new structure learning approach is proposed in limited data settings. In these settings, existing constraint-based causal discovery approaches struggle with statistical tests of independence. The main insight is that we can define an equivalence class of graphs based on an upper bound on the size of the conditioning set. Using this, a new learning algorithm is presented to learn the equivalence class.

**Strengths:**

The paper is well-motivated and clear in its contributions. The problem of causal discovery from limited datasets seems to be a significant one. The novel contributions include introducing the formalism of K-Markov equivalence which can then be used to derive a learning algorithm. The soundness of the algorithm is guaranteed and the completeness is shown in some cases. Overall the theoretical contributions seem strong in this work.

**Weaknesses:**

On the empirical side, comparisons are performed with state of the art structure learners on synthetic and semi-synthetic datasets. Since the motivation of the work was causal discovery from limited data, using some more real-world cases could have made the experiments much stronger. Overall, perhaps the empirical evaluation is not as strong as the theoretical contributions of the paper.


**Questions:**

Are there real-world examples of learning from limited conditioning sets that could perhaps be used to strengthen the experimental evaluation?

**Limitations:**

There are no limitations that are explicitly mentioned in the paper.

---

> ### Author Rebuttal · Authors · 2023-08-06
>
> We would like to thank the reviewer for their comments and feedback.
>
> "real-world examples of learning from limited conditioning sets"\
> We share the reviewer's sentiment that more experiments with real-world datasets would make the paper stronger. bnlearn repository is the main resource that causality researchers typically use, which is what we used as well. Note that the lack of datasets with ground truth causal structures is a common struggle with causality research, which makes the theoretical groundings of any proposed solution ever more important than some other fields. We hope our work to initiate a line of research and that the follow-up work could apply our methods to more datasets as these become available. We will make our code available upon publication to help speed up this process.
>
> A related practical setting where it is naturally difficult to condition on certain subsets of variables is the following: Consider a causal graph where some variables are high-dimensional, such as images, while others are binary such as labels or classifier outputs. It is very difficult to run reliable conditional independence tests with images in the conditioning set in practice. What can we learn about the causal structure then? An important future work is to extend our results to when an arbitrary set of CI tests cannot be performed. We hope that our work can motivate research in this direction.
>
> Once again, thank you for your positive comments! We would be happy to discuss these points further.

---

> > ### Comment · Reviewer_Gg3j · 2023-08-18
> > **Thanks**
> >
> > Thanks for the response. I can understand the difficulty with "real-world" causal structures, but the availability of code should help follow-on work.

---

### Official Review · Reviewer_iDG2 · 2023-07-06

**Soundness:** 3 good
**Presentation:** 4 excellent
**Contribution:** 4 excellent
**Rating:** 7
**Confidence:** 4

**Summary:**

The authors study the problem of causal learning with bounded conditioning sets. It is an important question that up to what set of equivalence graphs a learner can  infer a causal graph when it can only perform conditional independence tests with limited size of the conditioning sets. They characterize the equivalence set and propose a sound constraint-based algorithm for learning it.

**Strengths:**

The authors study an important research question. The paper is well-organized.

The results are quite relevant and important. It nicely establishes the connections between the existing results about Markov equivalence class and newly define k- Markov equivalence class that is the set of causal graphs that have the same conditional independences with conditioning sets at most k.

This results can be a foundation for further developments in causal graph learning.

**Weaknesses:**

A discussion about the complexity of the proposed algorithm will improve the paper.
For instance, what is the complexity of checking whether two given DAGs are k-Markov equivalent?

**Questions:**

See above.

**Limitations:**

 The authors adequately addressed the limitations.

---

> ### Author Rebuttal · Authors · 2023-08-06
>
> We would like to thank the reviewer for their constructive comments and we are happy to hear that they found our results relevant and important.
>
> "Complexity of checking k-Markov equivalence."
> This is a good question. Currently, one needs to explicitly construct k-closure graphs given the two DAGs and check whether they are Markov equivalent. The step to make two variables adjacent requires one to loop through all conditioning sets of size at most $k$. This would take $\mathcal{O}(n^k)$. This is the main time-consuming step. Afterward, one can test Markov equivalence using the existing approaches. For example, Hu and Evans 2020 show that one can do this in $\mathcal{O}(ne^2+n^2e)$ time, where $e$ is the number of edges. Thus, the overall algorithm will indeed be polynomial-time when $k=\mathcal{O}(1)$.
>
> The complexity of the learning algorithm will be similar to an early-stopped version of FCI, called Anytime FCI by Spirtes 2001. Although this is more complicated and would depend on other parameters in the graph, such as primitive inducing paths, and thus the number and location of unobserved confounders, we can also roughly bound this by $\mathcal{O}(n^{k+2})$ since for any pair, we will not be searching for separating sets beyond the $\mathcal{O}(n^k)$ subsets of size at most $k$. Further runtime improvements, such as RFCI by Colombo et al. 2012 might be possible, but this requires a further understanding of the structure of k-closure graphs, which might be an interesting future direction.
>
> We will add this discussion with the relevant citations below to the manuscript. Thank you for your time!
>
> [Hu and Evans] Zhongyi Hu, Robin Evans, "Faster algorithms for Markov equivalence," 2020.
>
> [Spirtes] "An Anytime Algorithm for Causal Inference," 2001.
>
> [Colombo, Maathuis, Kalisch and Richardson] "Learning High-Dimensional Directed Acyclic Graphs With Latent And Selection Variables," 2012.

---

### Official Review · Reviewer_fQhL · 2023-07-07

**Soundness:** 3 good
**Presentation:** 3 good
**Contribution:** 2 fair
**Rating:** 5
**Confidence:** 2

**Summary:**

While I have the background to understand this paper, I'm afraid I am not well positioned to judge its novelty and significance in the causal discovery subfield. In short, the PC algorithm struggle when conditioning sets are large, so the authors propose a modification of Markov equivalence for bounded conditioning sets, then graphically characterize this relationship, and then propose the k-PC algorithm that is more successful in synthetic small small regimes.

 Nothing in the math or presentation seems incorrect to me, though I did not closely examine the more technical parts of the paper. The experiments are on synthetic data, which is typical in this subfield, but that makes it difficult to know whether k-PC has practical advantages over PC in real-world cases, as there are likely regimes where PC is superior.


Overall, this seems like competent well done work, but I lean towards rejection for this paper. The modification of PC and Markov equivalence for bounded conditioning sets seems like a straightforward and simple modification of existing definitions, so the contribution does not seem novel and significant to me. If there were convincing empirical results that showed this change makes causal discovery viable in new real-world contexts I might be convinced, but as it stands its unclear to me whether the synthetic experiments are good evidence of this, especially with how little prose was in the main text describing them. This might be more appropriate for a conference or workshop that is more narrowly focused on causality.

I'm perfectly happy to be convinced otherwise, or have my opinion ignored in favor of reviewers/metareviewers with a better understanding of the subfield!



**Strengths:**

See main

**Weaknesses:**

See main

**Questions:**

See main

**Limitations:**

See main

---

> ### Author Rebuttal · Authors · 2023-08-03
>
> We would like to thank you for your feedback and insights. Our responses are below.
>
> "this seems like competent well done work"
> Thank you! We appreciate this.
>
> "The modification of PC and Markov equivalence for bounded conditioning sets seems like a straightforward and simple modification of existing definitions, so the contribution does not seem novel and significant to me."
> We respectfully disagree with this assessment. As we stated in the paper, by giving up conditioning on a set of variables, we no longer have the classic Markov equivalence condition. In fact, it is not a priori clear if there even is a simple graphical Markov equivalence condition when we restrict the size of conditioning sets.
>
> Our results show that one can do this by using ancestral graph machinery - which is typically used for handling latent confounders - in the context of characterizing k-Markov equivalence class for systems without latents. This, in our opinion, is a surprising finding. The comparison with LOCI shows the power this representation has over the existing work addressing this problem.
>
> In terms of novelty, the key lemmas are all derived from first principles, and as far as we are aware of, the k-Markov equivalence characterization is not implied by any of the existing literature.
>
> In terms of significance, our goal was to demonstrate that the proposed approach could lead to more robust causal discovery (various F1 scores are used in the experiments) in the small sample regime where large CI tests become unreliable. We believe that rendering causal discovery more reliable in practice is a significant goal and hope that our contribution takes us one step closer to this goal.
>
> We hope that this re-iteration of our contributions helps convince the reviewer that our results are novel and significant and that the reviewer would reconsider their evaluation in light of this.
>
> If you have follow-up questions, please let us know. We would be happy to engage further.
>
> Thank you once again for your time and feedback!

---

> > ### Comment · Reviewer_fQhL · 2023-08-15
> >
> > Thanks for the explanation! In the end I'm happy to see this accepted

---

### Official Review · Reviewer_XPAW · 2023-07-08

**Soundness:** 4 excellent
**Presentation:** 4 excellent
**Contribution:** 2 fair
**Rating:** 5
**Confidence:** 4

**Summary:**

This paper aims to address the problem that in constraint-based causal discovery based on conditional independence tests, the tests become statistically unreliable with limited data when the conditioning sets are large. The proposed solution is to use only conditional independence tests for relatively small conditioning sets. To facilitate this goal, this paper provides a characterisation of k-Markov equivalence between causal DAGs, in the sense of entailing the same conditional independence statements in which the conditioning set has a cardinality not exceeding k. Since the characterisation is in terms of maximal ancestral graphs, a FCI-like algorithm is proposed to learn such equivalence classes and is empirically evaluated against some benchmarks.

**Strengths:**

This paper is well written and very readable. The technical results are sound and the empirical evaluation is reasonable.

**Weaknesses:**

Unfortunately, as far as I can tell, this paper largely reproduces the results of an old paper by Peter Spirtes (2001), "An Anytime Algorithm for Causal Inference", Proceedings of the Eighth International Workshop on Artificial Intelligence and Statistics. It seems to me that all the main results and insights of this paper were already presented in Spirtes's paper. I may have missed something important. I will increase my score if the authors can convince me that their contributions remain sufficiently original and significant given Spirtes's paper.

**Questions:**

Are the main results of this paper a reproduction of Spirtes's (2001) results?

---

> ### Author Rebuttal · Authors · 2023-08-02
>
> We would like to first thank the reviewer for pointing out such a relevant paper to us. Please allow us to compare our work with Spirtes (2001).
>
> The short answer to your question "Are the main results of this paper a reproduction of Spirtes's (2001) results?" is no, they are not a reproduction of Spirtes's results. We elaborate below:
>
> First, Spirtes 2001 says: You can halt FCI after running it until any conditioning set of size k. The result will still be a correct PAG, but a potentially less informative one.
>
> There are at least three main algorithmic differences:
> 1. We are not trying to learn a PAG. Our goal post is different. Namely, we don't actually have any latents in the system, nor any selection bias. We define the equivalence class to be learned by the algorithm a priori, knowing that we won't be conditioning on sets of size greater than k given a DAG.
>
> 2. The undirected edges in our representation do not represent selection bias. We use them to represent a different graph union operation. The lack of latents gives us this flexibility, which allows us to distinguish different sets of graphs by using both - and o-o edge marks for this. Please see Definition 3.15 for how each edge in our k-essential graph represents something specific about our equivalence class of graphs without latents. This machinery allows us to develop a finer equivalence class representation than LOCI [20].
>
> 3. Even if we simply compare the two algorithms ignoring this context, since our goal post is different, we have additional orientation rules that do not appear in FCI. Please see the new rules R11 and R12 in Algorithm 1. This again is because our goal post is different from learning a PAG of the true causal graph. Furthermore, we modified the existing FCI rules since we do not have undirected edges that represent selection bias, and therefore our undirected edges are treated as if they are circle edges. This can be seen in lines 1064-1072 in the supplementary material.
>
> In terms of the contributions of the paper, we also have a new representation to capture all the d-separation statements of bounded conditioning set size. Importantly, we use these to give necessary and sufficient graphical k-Markov equivalence conditions between two causal DAGs. These are not implied by the results of Spirtes 2001, which focuses on the soundness of early stopping of FCI in relation to the true PAG.
>
> We hope that this convinces the reviewer that the results of our paper are not simply reproductions of Spirtes 2001. It goes without saying that we will cite this paper and thanks to your input, we will be able to put our paper's contributions in better and clearer context in relation to the existing causal discovery literature.
>
> Thank you once again for your valuable feedback.

---

> > ### Comment · Reviewer_XPAW · 2023-08-21
> > **Score increased**
> >
> > Thanks for the helpful response to my question. I still think the results are fairly easy given what is already shown by Spirtes, but I agree that some of them are not simply corollaries of Spirtes's results. I have increased my score to 5.

---

> > > ### Author Response · Authors · 2023-08-21
> > >
> > > Thank you very much for your reply and for your reconsideration. Based on your remark, in addition to citing and explaining this work in the main paper, we will also add a dedicated subsection in the Appendix to more closely compare the two papers. Specifically, for the causally sufficient case, we will add causal graphs to showcase how much causal knowledge can be extracted by our algorithm compared to Anytime FCI. We explain one such difference below:
> > >
> > > One example we may use for this comparison is the graph in Figure 2. Here, our algorithm will learn the representation on the right: We can infer that either $a$ causes $d$ or $d$ causes $a$, noted by the undirected edge $a$ --- $d$ in our representation, rather than the circle edge $a$ o---o $d$, which would allow $a$, $d$ to be non-adjacent in some DAGs. Anytime FCI here will instead output circle edges between $a$ and $d$, i.e., $a$ o--o $d$. Even if we try to incorporate the knowledge that the underlying graph is causally sufficient, it is not obvious how one can conclude that $a$ and $d$ must be adjacent in *every DAG that induces the same degree-$k$ d-separation statements* from the Anytime FCI output. One way to do this would be to remove the edge from this PAG and check whether $a$, $d$ can now be d-separated by some conditioning set of size at most $k$. If yes, this changes the equivalence class, which means the edge must be there in any DAG. Our local rule R12 can instead orient this as an undirected edge, signifying that $a$ and $d$ must be adjacent in every underlying DAG by leveraging our equivalence class representation, without having the run this type of post-processing which might be computationally intensive for larger graphs.
> > >
> > > We once again thank the reviewer for pointing us to Spirtes 2001 and enabling this discussion.

---

### Decision · Program_Chairs · 2023-09-21

**Decision:**

Accept (poster)

**Comment:**

The paper extends the PC algorithm in the small data regime, by limiting the size of the variable sets considered in the conditional independence tests. This goal is very relevant for practitioners, and the reviewers appreciate the fact that the proposed approach is soundly built on a definition of Markov equivalence, coming with necessary and sufficient equivalence conditions between two causal DAGs.

The rebuttal convincingly established the originality of the approach w.r.t. Anytime FCI (Spirtes 2001).  It is recommended that the discussion about the computational complexity be added to the camera-ready version.